



# ICON simulations of cloud diabatic processes in the warm conveyor belt of North Atlantic cyclone Vladiana

Anubhav Choudhary[1,*] and Aiko Voigt[2]

[1]Institute of Meteorology and Climate Research, Department Troposphere Research, Karlsruhe Institute of Technology, Karlsruhe, Germany
[2]Department of Meteorology and Geophysics, University of Vienna, Austria

**Correspondence:** Anubhav Choudhary (anubhav.choudhary@kit.edu)

**Abstract.** Warm conveyor belts are important features of extratropical cyclones and are characterized by active diabatic processes. Previous studies reported that the simulation of extratropical cyclones can be strongly impacted by horizontal model resolution. Here, we study to what extent and in which manner simulations of warm conveyor belts are impacted by model resolution. To this end we investigate the warm conveyor belt of the North Atlantic cyclone Vladiana that occured around 23 September 2016 and was observed as part of the North Atlantic Waveguide and Downstream Impact Experiment. We analyze a total of 18 limited-area simulations with the ICOsahedral Nonhydrostatic model run over the North Atlantic that cover a range of horizontal resolutions from 80 to 2.5 km, including the resolution of current low-resolution global climate models with parametrized convection as well as the resolution of future storm-resolving climate models with explicit convection. The simulations also test the sensitivity with respect to the representation of convection and cloud microphysics. With higher resolution, the number of WCB trajectories increases systematically and a new class of anticyclonic trajectories emerges that is absent at the lowest resolution of 80 km. WCB trajectories ascend faster and higher as resolution is increased. Explicitly resolving convection increases these changes further. We also diagnose the impact of increased resolution on the ascent velocity and vorticity of WCB air parcels and the diabatic heating that these parcels experience. With increasing resolution, ascent velocity increases at all pressure levels by around a factor of 3 and vorticity increases similarly strongly in the lower and middle troposphere. We find a corresponding increase in diabatic heating as resolution is refined, arising mainly from cloud-associated phase changes of water. Besides resolution, the treatment of convection has a stronger impact than the treatment of cloud microphysics in our simulations. We find no clear connection between the strength of diabatic heating within the WCB on the one hand and the deepening of cyclone Vladiana in terms of its central pressure on the other hand. An analysis of the pressure tendency equation shows that this is because diabatic heating plays a minor role for the deepening of the cyclone, which is dominated by temperature advection

## 1 Introduction

Diabatic processes play an important role for extratropical cyclones (Stoelinga, 1996; Wernli and Davies, 1997). In particular, latent heating from phase changes of water impacts the strength of cyclones (Booth et al., 2013). Most of the diabatic processes occur within coherent streams of ascending air known as warm conveyor belts (WCBs) (Harrold, 1973; Eckhardt et al., 2004).





Therefore, a realistic representation of WCBs in models and the diabatic processes within them is crucial for accurate predictions of extratropical cyclones at the weather time scale, and might also be needed in climate models for adequate simulations of extratropical cyclones and their response to climate change.

WCBs originate in the boundary layer of the cyclones' warm sector and ascend poleward, moving ahead of the cold front (Carlson, 1980; Joos and Wernli, 2012). During their cross-isentropic ascent to the upper troposphere, they are associated

with cloud formation and generate precipitation (Browning, 1990; Madonna et al., 2014; Pfahl et al., 2014). WCBs play a key role for the vertical transport of heat, moisture and atmospheric tracers (Stohl, 2001). Strong diabatic processes occurring within WCBs can have a strong influence on potential vorticity in the lower and upper troposphere, impacting the evolution of cyclones, their large-scale environment and blocking events (e.g., Wernli and Davies, 1997; Grams et al., 2011; Madonna et al., 2014; Binder et al., 2016; Joos and Forbes, 2016; Pfahl et al., 2015). Recent studies have also found convective activity

embedded within WCBs, leading to rapid vertical ascent of air parcels and intensified localized diabatic heating that further modifies potential vorticity and cyclone strength (Martinez-Alvarado et al., 2014; Binder et al., 2016; Rasp et al., 2016; Oertel et al., 2019 and Oertel et al., 2020). Joos, 2019 investigated the link between the top-of-atmosphere cloud-radiative effects and WCBs and highlighted how WCBs modulate the extratropical radiation budget.

Despite decades of model development, biases in climate models and differences between model projections of future cli-

mates remain substantial. Model limitations result predominantly from a parametrization 'deadlock', in particular because of small-scale cloud processes in the atmosphere (Randall et al., 2003; Jakob, 2010; Palmer and Stevens, 2019). This hinders the development of regional adaptation strategies to global climate change. Acknowledging the limitations of coarse-resolution global models and given the long history of unsuccessful attempts to solve the convection parametrization challenge, modeling centers around the world have started to develop storm-resolving models at the global scale in which horizontal model res-

olution is increased to a few kilometers so that the most rigorous aspects of deep convective motions in the atmosphere can be calculated directly and the parametrization for deep convection can be turned off (Satoh et al., 2019; Stevens et al., 2019; Stevens et al., 2020). By increasing resolution and treating deep convection in an explicit manner, it is hoped and in fact often reported that simulations of climate improve. For example, Senf et al., 2020 found that in the ICON model increasing resolution to storm-resolving scales of 2.5 km and representing deep convection explicitly leads to marked improvements in simulated

top-of-atmosphere cloud-radiative effects over the North Atlantic. Vergara-Temprado et al., 2020 found notable improvements in precipitation and the diurnal cycle for year-long simulations of European climate in high-resolution models with explicit deep convection.

One has reason to hope that the extratropical circulation improves in a similar manner in storm-resolving models. Model simulations of extratropical cyclones have often reported a strong sensitivity with respect to horizontal resolution (Champion

et al., 2011; Jung et al., 2006; Willison et al., 2015). In particular, a higher resolution typically leads to more intense cyclones (e.g., Chang and Fu, 2003; Jung et al., 2006; Colle et al., 2013; Eichler et al., 2013). Willison et al., 2013 found that the resolution sensitivity arises from a positive feedback between latent heating and cyclone strength, indicating that an inaccurate representation of moist processes and their associated latent heating can significantly affect simulations of storm tracks and the larger-scale circulation of the extratropics. This is also of concern when simulating the future climate, as in a warmer





atmosphere the combined effects of altered meridional temperature gradients and mesoscale latent heating complicate the warming response of extratropical storm tracks (Ulbrich et al., 2008 and Ulbrich et al., 2009).

With global storm-resolving models coming into application, we find it important to understand how increasing horizontal resolution affects model simulations of extratropical cyclones, WCBs and the diabatic processess associated with them. Although previous studies have addressed the impact of resolution on cyclones, we are not aware of a systematic study of the

impact of resolution on the simulation of WCBs and the link to cyclone intensity at the synoptic scale. Here we address this question by means of a cyclone case study from the NAWDEX field campaign (Schäfler et al., 2018). We study a cyclone named Vladiana that occurred during 22-25 September, 2016, over the North Atlantic and whose WCB was well developed (Oertel et al., 2019; Oertel et al., 2020). We analyze a suite of simulations of Vladiana with the ICOsahedral Nonhydrostatic model (ICON; Zängl et al., 2015) in limited-area setup over a large North Atlantic domain at six horizontal resolutions rang-

ing from 80 to 2.5 km. The simulations are performed with 1-moment and 2-moment bulk cloud microphysics. The higher resolutions of 10, 5 and 2.5 km are performed with parametrized as well as explicit convection.

We address the following questions:

1. How do horizontal resolution, the treatment of convection and the treatment of cloud microphysics affect the simulation of the WCB associated with cyclone Vladiana?

2. How sensitive is diabatic heating within the WCB to these modeling choices?

3. Do the sensitivities of the WCB diabatic processes affect the deepening of cyclone Vladiana?

The paper is organized in the following order. Section 2 describes the model simulation and analysis methods. This is followed by an analysis of the WCB and diabatic processes in Section 3, and an analysis of the impact of diabatic processes on the deepening of the cyclone by means of the pressure tendency equation in Section 4. The paper concludes with a summary

of the main findings in Section 5.

## 2 Method

### 2.1 Model simulations

We consider simulations of the North Atlantic extratropical cyclone Vladiana. Vladiana occured during the NAWDEX field campaign in fall 2016 (Schäfler et al., 2018) and exhibited a pronounced WCB. Oertel et al., 2019 and Oertel et al., 2020

studied this case using the COSMO model to understand the convective processes embedded within the WCB and their impact on the larger-scale circulation. Here, we use the atmospheric component of the ICON modeling system to study in detail the diabatic processes within the WCB as represented by ICON.

We apply ICON version 2.1.00 in a limited-area set-up with the physics package for numerical weather prediction (Zängl et al., 2015). The setup is described in detail in Stevens et al., 2020 and Senf et al., 2020 and largely follows the tropical

Atlantic setup of Klocke et al., 2017. The simulations are initialized from and forced at their lateral boundary with analysis and





forecast data from the ECMWF-IFS Integrated Forecasting System. Six different horizontal resolutions are considered: 80, 40, 20, 10, 5 and 2.5 km. 75 model levels are used. For the 80, 40 and 20 km simulations, convection is parametrized based on the mass flux schemes for shallow and deep convection of Tiedtke, 1989 and Bechtold et al., 2008. For the three finest resolutions of 10, 5 and 2.5 km we analyze simulations in which convection is parametrized as well as simulations in which convection is

represented explicitly, i.e., the deep and shallow convection schemes are disabled. The simulations with explicit convection are distinguished by "EC" in the following.

All simulations are run for both 1- and 2-moment cloud microphysics, which are based on Doms et al., 2005 and Seifert and Beheng, 2006, respectively. The 1-moment scheme includes the specific mass of water vapor, cloud water, rain water, cloud ice, snow and graupel, where graupel is relevant for the explicit simulation of deep convection (Baldauf et al., 2011).

The 2-moment scheme in addition includes the number concentration of the aforementioned hydrometeor species and further includes hail. The 1-moment scheme is used in operational forecasts of the German weather service DWD. The 2-moment scheme has been developed for high-resolution simulations with explicit convection. We here apply the 2-moment scheme also for coarse-resolution simulations with parametrized convection. Although this is not recommended (Prill et al., 2020), these simulations corroborate our finding that the treatment of cloud microphysics has a minor effect on our results.

We analyse a total of 18 simulations. The simulations cover a period of 4 days, starting from 2019-09-22, 0UTC to 2019-09-26, 0UTC. The simulation domain covers the North Atlantic as well as much of Europe and Northern Africa (78°W-40°E and 23°N-80°N; see Fig. 3 of Stevens et al., 2020). This ensures that the simulations include the entire temporal and spatial extent of Vladiana. The simulations analyzed here are a subset of those analyzed by Senf et al., 2020. For more details regarding the model setup, readers are thus referred to Senf et al., 2020, who evaluate the simulation in terms of clouds and top-of-atmosphere

cloud-radiative effects.

## 2.2 Synoptic development of cyclone Vladiana

Fig. 1 provides an overview of the simulated synoptic evolution of cyclone Vladiana based on the 2.5 km-EC simulation with 1-moment cloud microphysics. Vladiana intensified during 22 and 23 September, 2016. In the beginning, at 12 UTC on 22 Sep, an upper-level positive PV anomaly occurred around 60°N in a strong baroclinic zone. 24 hours later, at 12 UTC on 23

September, the cyclone had intensified and deepened to a minimum sea level pressure of below 980 hPa (exact values depend on the model setup; see Fig. 9). At this time, the upper level PV distribution formed a strong elongated ridge that was aligned with the warm front of the cyclone, with low PV values north of the British Isles. Distinct surface cold and warm fronts during this time were evident in the 850 hPa equivalent potential temperature field (Fig. 1b). In the next 24 hours, the cyclone decayed while moving further northward.

The simulated evolution agrees well with the synoptic evolution described by Oertel et al., 2019 based on ECMWF analysis and forecast data (their Fig. 2 g-i) and the PV evolution in the COSMO model described by Oertel et al., 2020 (their Fig. 2 d-f). Note that the color scales and contour levels in our Fig. 1 are chosen as in Oertel et al., 2019 and Oertel et al., 2020 for better comparison. Moreover, Senf et al., 2020 showed that the simulated spatial pattern of the cloud band associated with Vladiana's



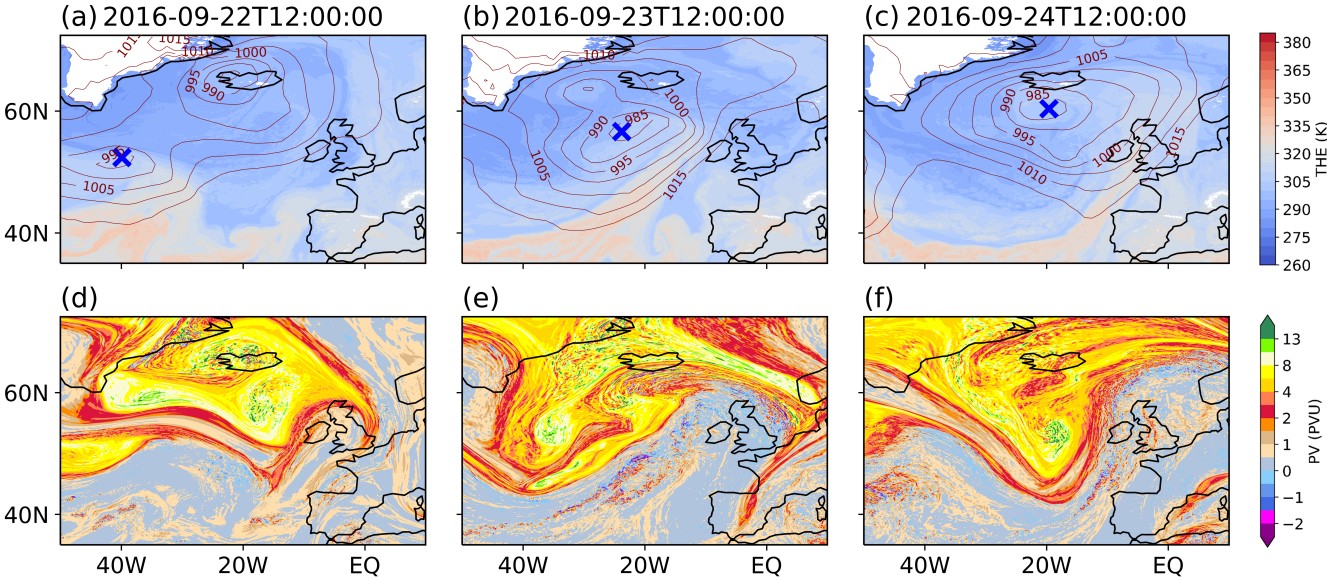

**Figure 1.** Synoptic evolution during the lifetime of cyclone Vladiana at (a, d) 22 September 2016 12 UTC, (b, e) 23 September 2016 12 UTC and (c, f) 24 September 2016 12 UTC. (a-c) Equivalent potential temperature (THE) at 850 hPa (colour shading) and mean sea level pressure (contour lines, units of hPa). The cyclone position as given by the minimum sea level pressure is shown by the blue cross. (d-f) Potential vorticity (PV) on the 320 K isentrope. The figure is based on the 2.5 km-EC simulation with 1-moment cloud microphysics.

WCB compares well with satellite observations (their Fig. 2). In summary, the simulations analyzed here capture the overall

evolution of cyclone Vladiana and the associated cloud fields.

## 2.3 Computation of WCB trajectories

To investigate the diabatic processes occurring within the WCB, we perform Lagrangian trajectory analyses for all 18 simulations using the LAGRANTO tool (Sprenger and Wernli, 2015) and hourly model output. LAGRANTO requires the input data to be on a regular latitude-longitude grid. We remap the model output from the ICON triangular grid to a regular latitude-longitude

grid using conservative remapping as implemented in the Climate Data Operators (Schulzweida, 2019). The resolution of the regular grid corresponds to the resolution of the corresponding ICON grid. For the 80, 40, 20, 10, 5 and 2.5 km simulations, the model output is remapped to regular grids with a longitudinal and latitudinal spacing of 0.8, 0.4, 0.2, 0.1, 0.05 and 0.025 degrees, respectively. Based on remapped fields of wind and pressure, 48 hours forward running trajectories beginning from 22 September 2016 12 UTC are calculated. The trajectories are seeded at 14 equally-spaced pressure levels between 1050 hPa and

790 hPa and from every grid point in a predefined seeding region near the warm sector of the cyclone (45° W to 0° and 35° N to 60° N). The seeding region is based on the WCB starting positions identified from the ECMWF offline trajectories by Oertel et al., 2019 (see their Fig. 1). The seeding points are based on the 20 km simulation. For all resolutions the same seeding points are used. Because all simulations have the same number of seeding points, the number of trajectories can be compared across





resolutions. This approach also provides good sampling of the WCB while limiting the number of trajectories to a practicable

amount, especially for the highest resolution of 2.5 km. After the trajectories are calculated, the WCB trajectories are selected
as those with an ascent larger than 600 hPa within 48 hours (Wernli and Davies, 1997), and the variables of interest are traced
along the trajectories.

Two methodological choices should be pointed out. First, trajectories are only seeded once at the starting time step, i.e.,
there is only one trajectory starting from each seeding point. We found this to be sufficient to sample the evolution of the WCB

in time and space, as some trajectories ascend earlier and some trajectories ascend later in the considered 48-hour period. This
is illustrated in supplementary Fig. S1. Second, we use offline trajectories. This was necessary as the employed ICON version
does not include the capacity for online trajectories. As a result, and with a few exceptions at the highest resolutions and for
explicit convection, our trajectories represent slantwise ascent and are similar to offline trajectories based on ECMWF-IFS data
(Fig. 2 d-f in Oertel et al., 2019). Because slantwise-ascending trajectories represent the majority for Vladiana (Oertel et al.,

2019), we expect our analysis to sample the mean diabatic processes within the WCB in an adequate manner.

## 2.4 Diabatic heating

The atmospheric physics package of ICON contains various schemes to represent subgrid-scale diabatic processes and their
impact on the resolved circulation. A detailed description of the formulation of diabatic processes is provided in the ICON tu-
torial (Prill et al., 2020). For the purpose of our study, the temperature tendencies due to diabatic processes are most important,

i.e., the diabatic heating rates (DHR). In ICON, DHR result from microphysical processes (including saturation adjustment),
radiation interaction, turbulence, parametrized convection, horizontal diffusion as well as drag from subgrid-scale orography
and non-orographic gravity waves. In our simulations total DHR (hereafter, $\text{DHR}_{total}$) is diagnosed as well as its individual
components. DHR from horizontal diffusion and subgrid-scale orography and non-orographic gravity waves is found to be be
small and thus not shown separately. DHR from water phase changes, i.e., latent heating, can occur as part of the microphysics

scheme, most notably via the saturation adjustment as well as in the convection scheme (where it leads to convective precipi-
tation). Total DHR and its components are written out every 1 hour as instantaneous rates. Joos and Wernli, 2012 showed that
instantaneous values provide a good approximation of DHR accumulated over 1 hour.

Motivated by the work of Schäfer and Voigt, 2018 on the cloud-radiative impact on an idealized extratropical cyclone, we
also diagnose DHR from cloud-radiation interaction. Cloud-radiative heating is computed by means of all-sky and clear-sky

radiative fluxes as

$$DHR|_{crh} = \frac{1}{\rho c_v} \frac{\partial (F^{all} - F^{clr})}{\partial z},$$

where $\rho$ is air density, $c_v$ is the specific heat capacity of air at constant volume and F is the net radiative flux in all-sky (with
clouds) and clear-sky (without clouds) conditions, respectively. Clear-sky fluxes are diagnosed by an additional diagnostic
radiative transfer calculation with cloud fraction set to zero.





## 3 WCB trajectories

We first study WCB trajectories across model setups in terms of their number, subclasses and mean ascent characteristics in Section 3.1. We then study the evolution of diabatic heating along the trajectories in more detail in Section 3.2.

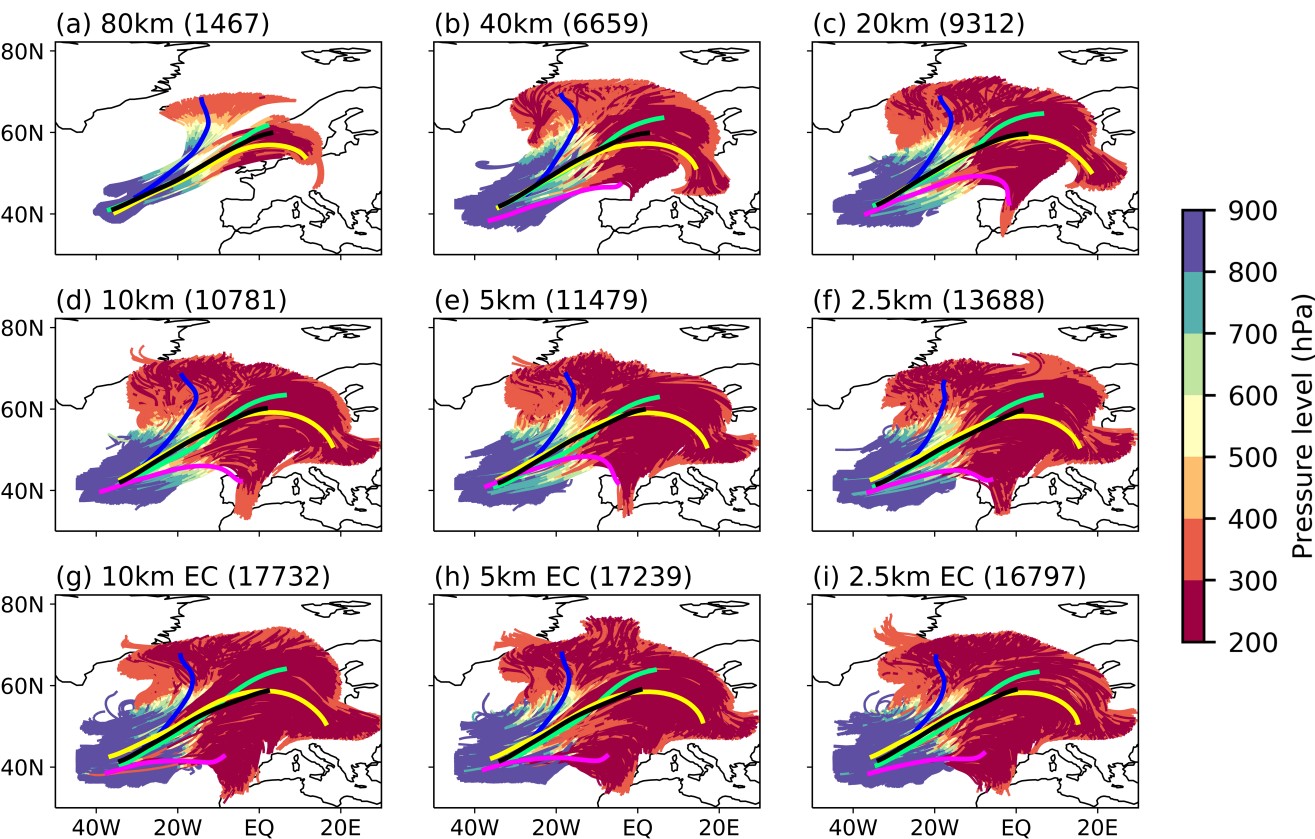

**Figure 2.** WCB trajectories identified from 48-hour forward trajectories in dependence of model resolution. The lower row (g-i) shows simulations for explicit convection (indicated by EC). The number in the bracket gives the number of WCB trajectories. Trajectories are coloured according to their pressure level. The thick coloured lines represent the mean path of different subclasses of trajectory, namely Trajectory 1 (blue), Trajectory 2 (green), Trajectory 3 (red), Trajectory 4 (magenta). The mean of all trajectories is shown in black. All simulations use the 1-moment cloud microphysics.

### 3.1 Number, subclasses and mean ascent properties

As described by Oertel et al., 2019, the ascending region of WCB was located in a region around 40 to 50° N and 40 to 10° W in the warm sector of the cyclone. This can be inferred from the maps of potential temperature (Fig. 1a-c) and WCB trajectories calculated for our ICON simulations (Figs. 2 and 3). Our simulations capture the multiple outflow branches of the WCB, i.e.,



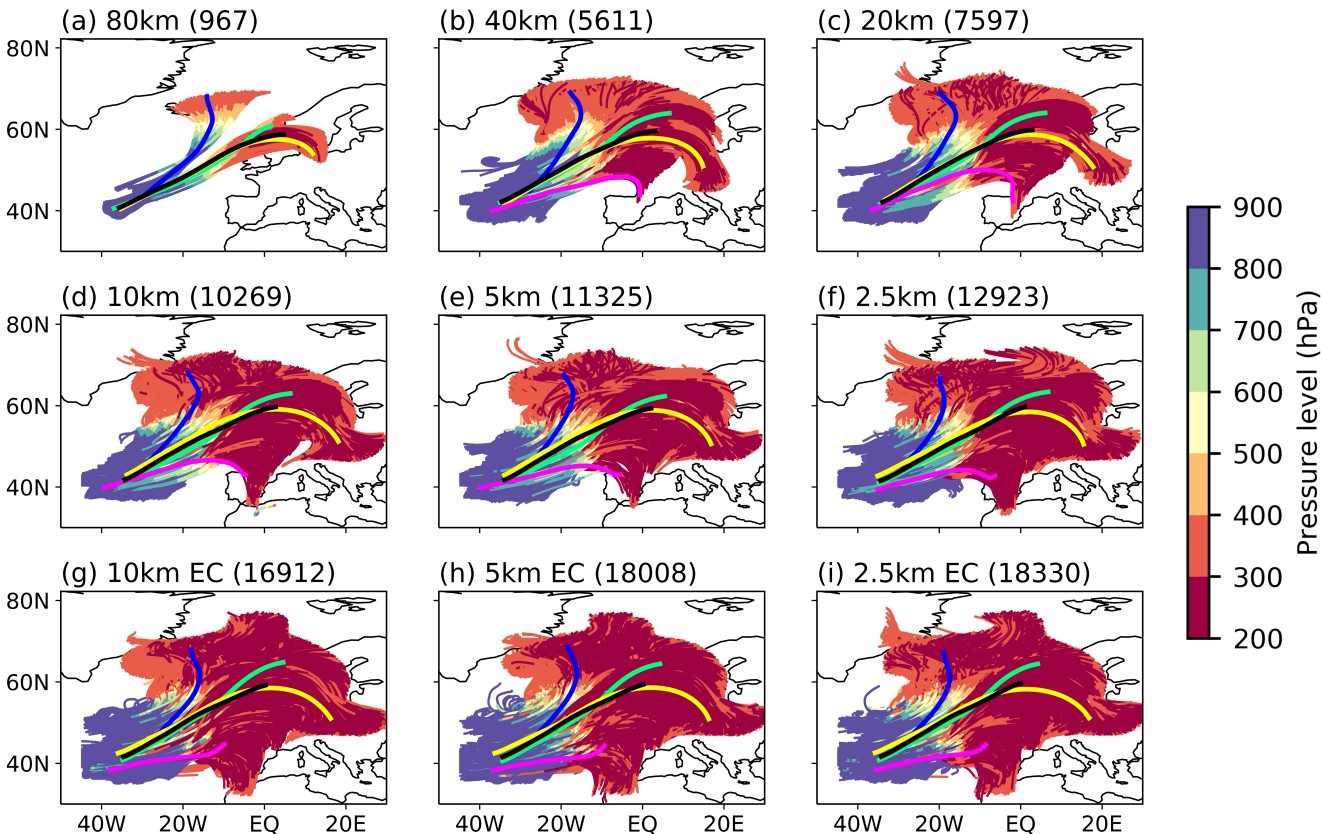

**Figure 3.** Same as Fig. 2 but for the simulations with 2-moment cloud microphysics.

its dichotomous nature (Martinez-Alvarado et al., 2014). Most WCB trajectories turn anticyclonically into the downstream upper-level ridge. A smaller fraction of the trajectories form a cyclonic branch that wraps around the centre of the cyclone.

Figs. 2 and 3 further study how the WCB trajectories change across model resolutions and for different treatments of convection and microphysics. The WCB strength – measured by the number of identified WCB trajectories – differs considerably across resolutions. The number of trajectories increases substantially as the resolution increase. In fact, for the finest resolution of 2.5 km, around 10 times more trajectories are identified than for the coarsest resolution of 80 km (Figs. 2 and 3, panels a-f). When convection is treated explicitly, the number of trajectories increases further, and is 50% higher compared to simulations that use the same resolution but parametrized convection (Figs. 2 and 3, panels g-i). For explicit convection the number of trajectories does not change much for resolutions between 10 and 2.5 km, indicating convergence with respect to model resolution. This convergence is not found when convection is parametrized. Comparing Figs. 2 and 3 shows that the treatment of microphysics has no substantial impact. Overall, we find that the WCB becomes more pronounced with finer resolution and explicit treatment of convection, while microphysics has no marked impact.



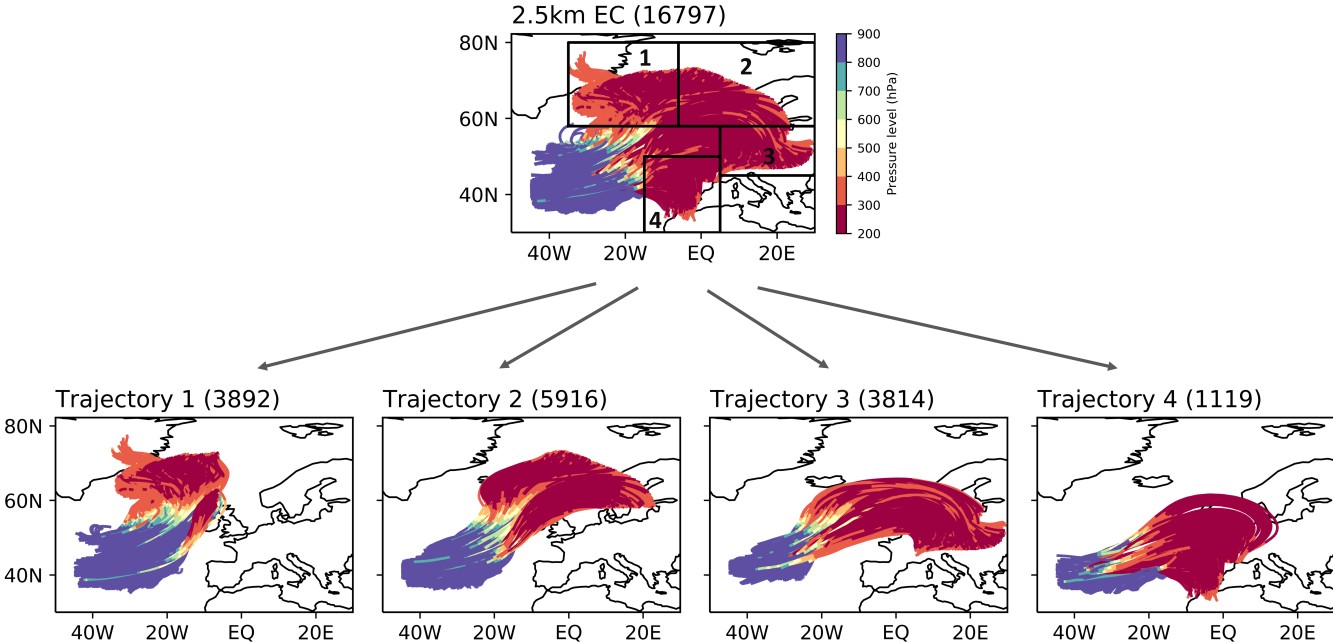

**Figure 4.** Separation of WCB trajectories into four subclasses. An example is shown here for the 2.5 km-EC simulation with 1-moment cloud microphysics. The trajectories are clustered based on their final location. The regions are defined as latitude-longitude boxes shown in the top figure. Their spatial extent is as follows. Trajectory 1: -35 – -6° E, 60 – 80° N; Trajectory 2: -6 – 30° E, 58 – 80° N; Trajectory 3: 5 – 30° E, 47 – 56° N; Trajectory 4: -20 – 0° E, 30 – 50° N.

Another finding from Figs. 2 and 3 is that the WCB consists of several subclasses of trajectories that differ in terms of their direction and ascent pattern. To investigate this further, we separate the WCB trajectories into four subclasses based on their final location. We refer to these subclasses as Trajectory 1, Trajectory 2, Trajectory 3 and Trajectory 4. Fig. 4 presents an example of the separation for the 2.5 km-EC simulation with 1-moment cloud microphysics. In Figs. 2 and 3, the mean trajectory of each subclass is included as a colored line. Trajectory 1 corresponds to the cyclonic branch of the WCB, wheras Trajectories 2, 3 and 4 belong to the anticyclonic branch.

Fig. 5 depicts the number of trajectories in each subclass as well as the total number of trajectories as a function of resolution. The number of trajectories for each subclass increases with increasing resolution and explicit treatment of convection, while the sensitivity with respect to the treatment of microphysics is weak. The largest contribution stems from the anticyclonically turning subclass Trajectory 2, which contributes about 50 percent to the total number of trajectories. The subclasses Trajectory 3 and Trajctory 1 contribute about equally. The subclass Trajectory 4 contributes relatively little and is absent for the coarsest resolution of 80 km.

We quantify the impact of resolution on the ascent dynamics of the WCB air parcels by analysing the main ascent period (MAP) in Fig. 6. MAP is defined as the period during which the actual ascent occurs, i.e., the time period between the minimum and maximum height (measured in pressure; Oertel et al., 2019). Panel a shows the mean MAP, while panel b shows the





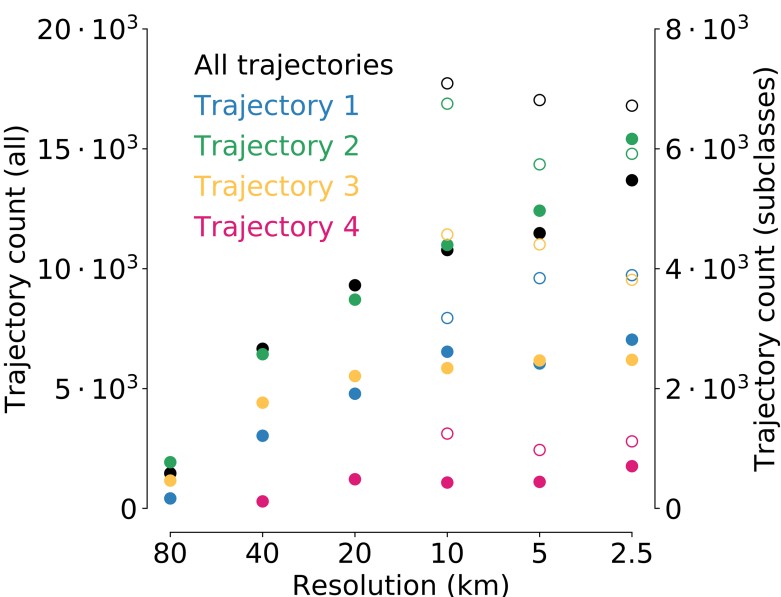

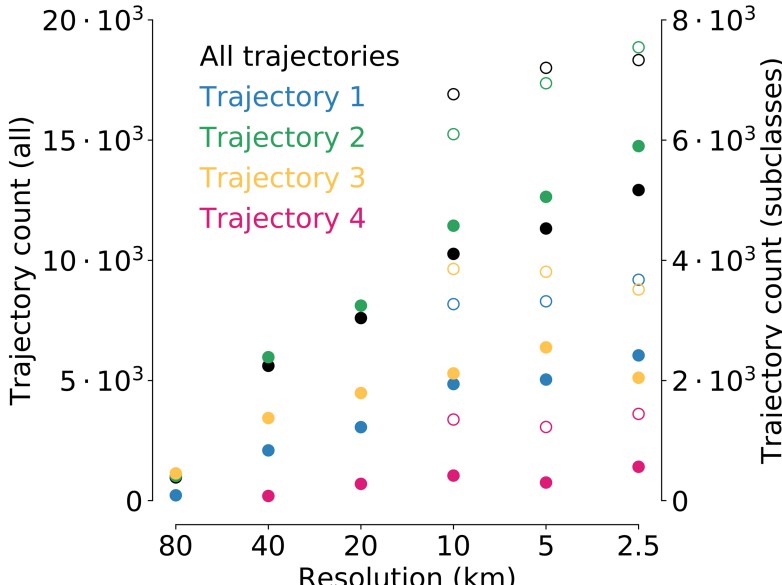

**Figure 5.** (a) Number of WCB trajectories as a function of horizontal resolution. The total number of trajectories is shown in black, with values given by the left y-axis. The number of trajectories in the 4 subclasses is shown in colors, with values given by the right x-axis. Filled and open markers correspond to parametrized and explicit convection, respectively. The (a) upper and (b) lower plots are for 1- and 2-moment cloud microphysics.




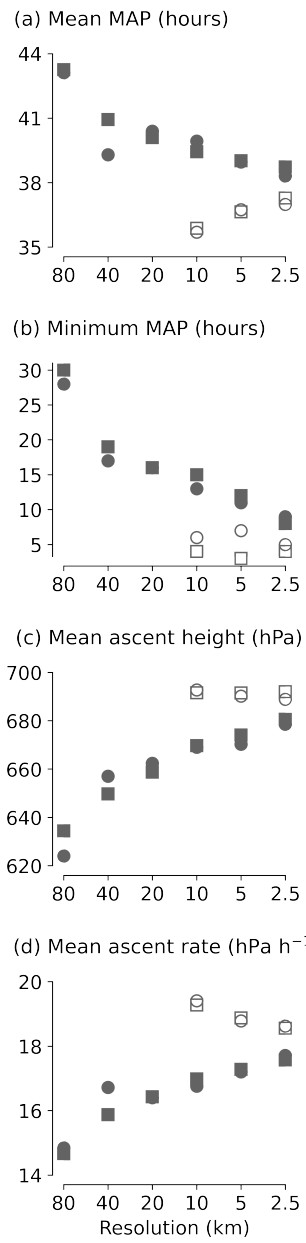

**Figure 6.** Statistics of the main ascent period (MAP) as a function of horizontal resolution. The filled and empty markers represent parametrized and explicit convection, respectively, while circle and square markers represent 1- and 2-moment cloud microphysics.





minimum MAP. We note that for all resolutions, the maximum MAP is 48 hours, as there always exist at least one trajectory

that takes the whole 48 hour period to complete its ascent. Panel c shows the mean ascent height. Panel d shows the mean ascent rate, which equals the ratio of mean ascent height and mean MAP.

As resolution is increased, parcels on average ascend faster and higher. For parametrized convection, the effect of increasing resolution is nearly linear: each increase in resolution by a factor of two leads to roughly the same reduction in mean MAP and increase in mean ascent height and rate. When comparing the finest and coarsest resolution, we find that at 2.5 km resolution the

ascent is 6 hours faster and 60 hPa higher than for the 80 km resolution. When convection is treated explicitly, the ascent occurs even faster and over a larger vertical distance. The effect of treating convection explicitly is largest for the 10 km resolution and smallest for the 2.5 km resolution. This is consistent with the expectation that the resolved circulation gradually replaces the convection scheme as the resolution increases, because of which the effect of the convection scheme should be smallest at the highest resolution. In contrast to the marked impact of model resolution and treatment of convection, we again find no

substantial impact of the treatment of microphysics. This can be seen by the close overlap of the circle and square symbols that distinguish the 1- and 2-moment cloud microphysics in Fig. 6. In summary, with increased resolution and explicit convection, the model simulates a quicker and higher ascent of WCB parcels.

Oertel et al., 2019 (their Table 1) considered online trajectories from COSMO simulations and offline trajectories from ECMWF-IFS data. The COSMO simulations were run with explicit deep but parametrized shallow convection and a resolution

of 2.2 km. Our ICON simulations at 2.5 km show slower ascent that reaches somewhat higher for explicit convection compared to the COSMO results of Oertel et al., 2019. The slower ascent is due to the fact that we use offline instead of online trajectories. The ECMWF-IFS data were obtained from simulations with parametrized convection and a horizontal resolution of 9 km. Our ICON simulation with parametrized convection and 10 km resolution agrees well with the ECMWF-IFS results of Oertel et al., 2019, with very similar values for mean and minimum MAP (39 vs. 40 hr; 13 hr vs. 13 hr), mean ascent height (653 vs.

669 hPa) and mean ascent rate (17 vs. 16.8 hPa/hr). Our results thus indicate that the differences between the ECMWF-IFS and COSMO trajectories analysed by Oertel et al., 2019 result both from differences in the model resolution and treatment of convection as well as from differences in the use of online versus offline trajectories.

## 3.2 Dynamics of parcel ascent and diabatic heating within the WCB

In this section, we study the dynamics of the WCB air parcels and the diabatic heating that they experience as a function of their

vertical position. Fig. 7 a depicts the parcels' pressure level as a function of time for the coarsest and finest resolution. While both resolutions exhibit a broadly similar evolution of air parcels, the spread between the trajectories is distinctively larger at 2.5 km resolution. The increased spread results from the fact that as resolution increases, the WCB trajectories become more diverse. This effect is illustrated in supplementary Figs.S1 and S2, which also show that treating convection explicitly further increases the diversity between WCB trajectories and that the ascent occurs in two main time periods. The latter is consistent

with Oertel et al., 2019.

Fig. 7 b-d illustrates the parcel dynamics as a function of their vertical location in terms of ascent velocity as well as absolute and potential vorticity. Consistent with the MAP mean ascent rate shown above in Fig. 6, the parcel ascent systematically



**Figure 7.** (a) Vertical location of WCB air parcels (measured by their pressure level) as a function of time for simulations with 80 km and 2.5 km resolution, parametrized convection and 1-moment cloud microphysics. The shading illustrates the spread between trajectories and is given by the $25^{th} - 75^{th}$ percentile. (b) ascent velocity, (c) absolute vorticity and (d) potential vorticity as a function of resolution and pressure level averaged over all trajectories. Lines with filled and empty markers represent simulations with parametrized and explicit convection, respectively. All simulations shown here use 1-moment cloud microphysics.





strengthens as resolution is increased, with the maximum ascent velocity shifting to lower levels (panel b). Absolute and potential vorticity display the expected vertical profile within a WCB, with maximum values in the lower troposphere. Similar
to ascent velocity, increasing resolution leads to a systematic increase in absolute and potential vorticity. For all three quantities, the simulations with explicit convection display the strongest ascent and vorticity. As a result, the maximum values for ascent and vorticity are about three times higher for the 2.5 km resolution with explicit convection than for the 80 km resolution with parametrized convection. Consistent with our results in Sect. 3.1, the treatment of microphysics has a minor impact.

Fig. 8 a characterizes the diabatic heating along WCB trajectories. Total diabatic heating systematically increases with res-
olution. In fact, between the 80 and the 2.5 km resolutions the peak diabatic heating increases by almost a factor of three. To quantify to what extent the increase in diabatic heating results from smaller-scale ascent and its correlation with diabatic heating, we recalculated the trajectories with all simulations interpolated onto the same 40 km grid. The systematic relationship between DHR and resolution still holds in this case as depicted in supplementary Fig. S3, although the resolution impact decreases to a factor of 2 between the 80 and 2.5 km simulation. This shows that the increase in diabatic heating arises not only from changes at smaller scales but to a large extent is due to changes on scales of 40 km and larger.

Total diabatic heating is dominated by microphysical processes, which exhibits almost the same vertical pattern as total diabatic heating (Fig. 8 b). The increase in microphysical heating likely reflects the stronger ascent and thus larger condensation when resolution is increased. Convection contributes in the lower troposphere, where it infact dominates total diabatic heating for the low-resolution simulations (Fig. 8 e). The contribution of convection decreases as resolution is increased, as is expected
because more and more of the vertical transport can be achieved by the resolved grid-scale circulation. The contribution by turbulence is relatively small and limited to the lower and middle troposphere (Fig. 8 f). The contribution by cloud-radiative and clear-sky radiative heating is negligible (Fig. 8 c and d). This is because within the WCB the air parcels are typically within clouds and not at the boundary between clear-sky and cloudy regions.

Overall, we find that the diabatic heating strongly intensifies with resolution. The increase in diabatic heating occurs in a
gradual manner, with no indication of significant structural changes with increased resolution. Diabatic heating and its resolution dependence is dominated by cloud microphysics.

## 4   Synoptic evolution of cyclone Vladiana and missing link to WCB diabatic processes

Previous work has shown that the diabatic processes occurring within WCBs can have a strong influence on the distribution of potential vorticity in the lower as well as upper troposphere, and hence on the evolution of midlatitude cyclones (e.g., Wernli
and Davies, 1997; Grams et al., 2011; Madonna et al., 2014; Binder et al., 2016; Joos and Forbes, 2016). In this section, we study to what extent the sensitivity of the WCB diabatic heating found in Sect. 3 imprints on the synoptic evolution of cyclone Vladiana across the different model setups.

We characterize the evolution of cyclone Vladiana by means of its central pressure at mean sea level, which is shown in Fig. 9. To remove possible spin up effects, the figure starts on 12 UTC of 22 Sep 2016, i.e., 12 hours after the model initialization.
The cyclone deepens and reaches its minimum central pressure at around 12 UTC on 23 Sep 2016. Although the deepening

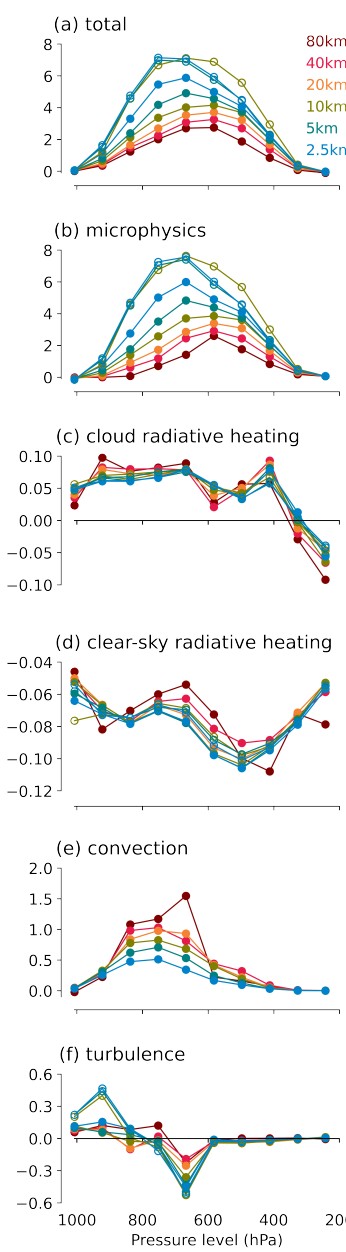

**Figure 8.** (a) Total diabatic heating rate along pressure levels for different resolutions calculated as mean over the trajectories. The lines with filled and empty markers represent simulations with parametrized and explicit convection, respectively. All simulations shown here use 1-moment cloud microphysics. (b-f) Same as (a) but for heating from individual processes.





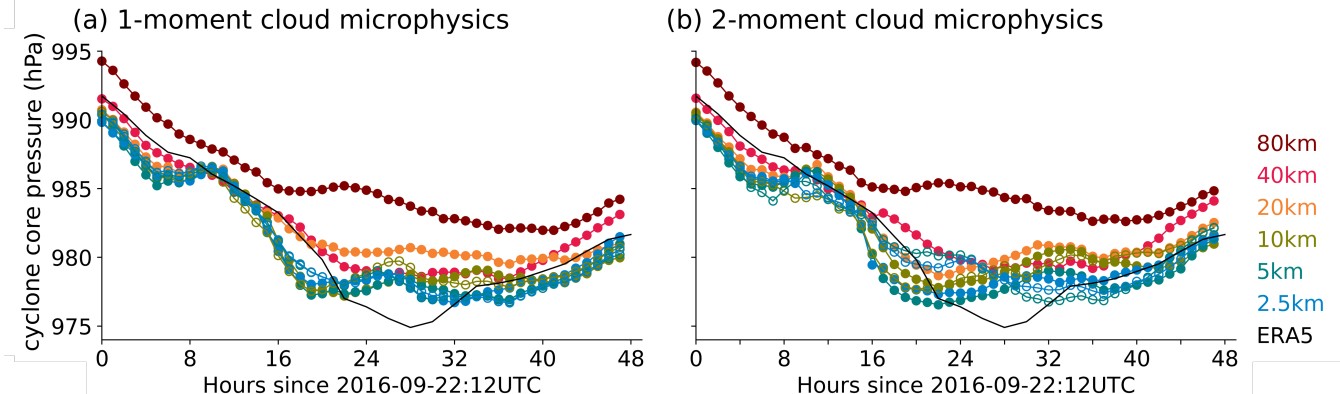

**Figure 9.** Central pressure of cyclone Vladiana since 22 September 2016 12 UTC for 1- and 2-moment cloud microphysics, respectively. Filled and empty markers distinguish simulations with parametrized and explicit convection. The central pressure derived from ERA5 is shown as the black thin line.

for the 80 km resolution is less pronounced than for the other resolutions, we overall find no systematic impact of model resolution or treatment of convection on the cyclone evolution. For example, for the 2-moment simulations at 2.5 km resolution the cyclone is comparably strong with parametrized convection but not for explicit convection.

Because the cyclone does not systematically strengthen with resolution, the cyclone strength and the magnitude of WCB
diabatic heating are not linked to each other. For example, although the WCB diabatic heating is strongest for the 10, 5 and 2.5 km simulations with explicit convection, the cyclone is not strongest in these simulations. This indicates that cyclone Vladiana is not impacted considerably by diabatic processes occurring in its associated WCB. We investigate this further in the next subsection by means of the surface pressure tendency equation.

**4.1   Surface pressure tendency equation (PTE)**

The surface pressure tendency equation quantifies the impact of advection and diabatic heating on the surface pressure evolution (Knippertz and Fink, 2008; Knippertz et al., 2009). This approach is commonly referred to as PTE, which is shorthand for pressure tendency equation. It can be applied to understand the processes driving the deepening of midlatitude cyclones. Following the work of Fink et al., 2012, we use it to understand to what extent cyclone Vladiana was affected by diabatic heating. For a detailed description of PTE analysis and its implementation, please refer to Fink et al., 2012 and Papavasileiou
et al., 2020.

The PTE is based on the equation for the local derivative of surface pressure,





$$\underbrace{\frac{\partial p_{\mathrm{sfc}}}{\partial t}}_{\mathrm{D}p} = \underbrace{\rho_{\mathrm{sfc}}\frac{\partial \phi_{100\mathrm{hPa}}}{\partial t}}_{\mathrm{D}\phi} + \underbrace{\rho_{\mathrm{sfc}}R_{\mathrm{d}}\int\limits_{\mathrm{sfc}}^{100\mathrm{hPa}}\frac{\partial T_{\mathrm{v}}}{\partial t}\,\mathrm{d}(\ln p)}_{\mathrm{ITT}}$$

$$+ \underbrace{g(E-P)}_{\mathrm{E-P}} + \mathrm{RES}_{\mathrm{PTE}}$$

where $p_{\mathrm{sfc}}$ and $p$ are surface pressure and atmospheric pressure, respectively, $\rho_{\mathrm{sfc}}$ is surface air density, $R_{\mathrm{d}}$ is the dry air gas constant, $\phi_{100\,\mathrm{hPa}}$ is the geopotential at 100 hPa and $T_{\mathrm{v}}$ is the virtual temperature. $g$ is the constant of gravitational acceleration.

$E$ and $P$ are surface evaporation and precipitation. $\mathrm{RES}_{\mathrm{PTE}}$ represents any residual in the analysis that can arise for example due to the spatiotemporal discretisation.

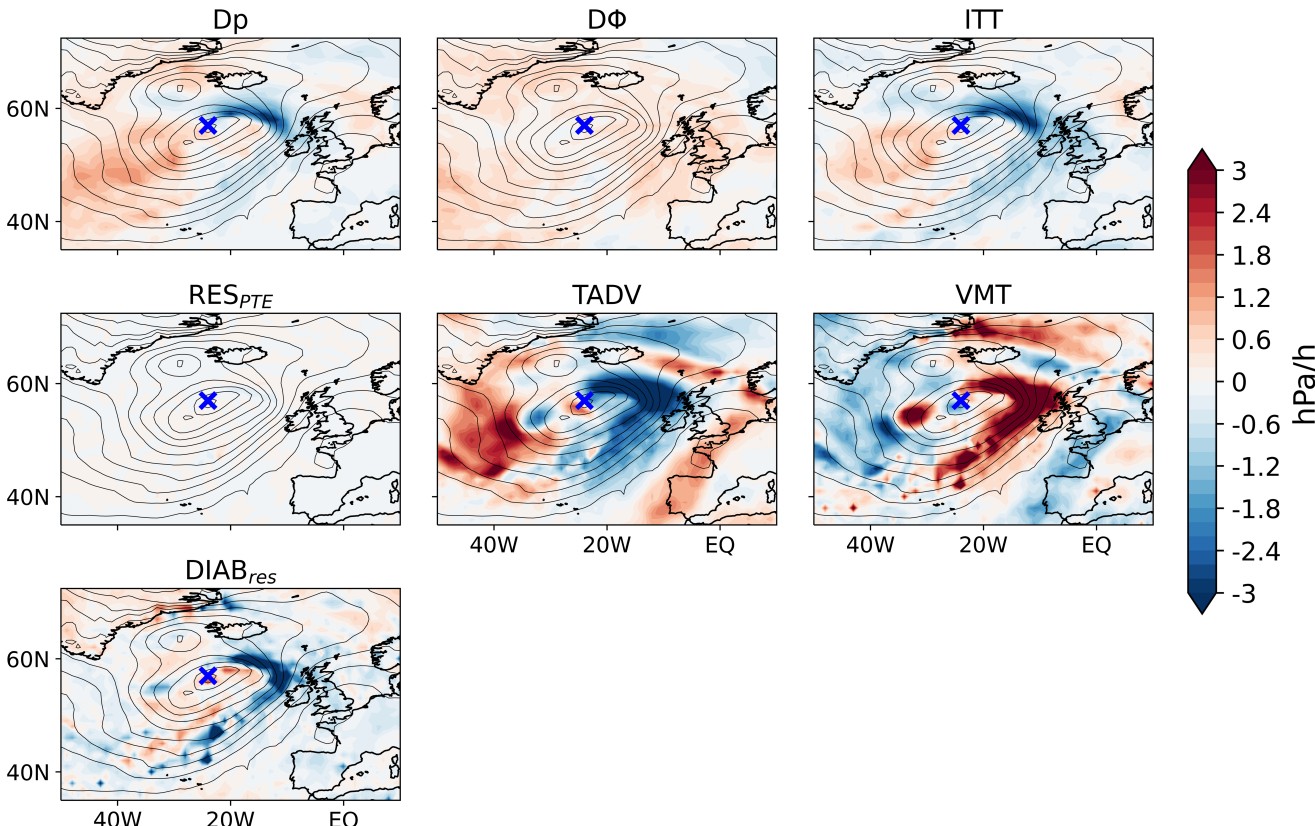

**Figure 10.** Illustration of the PTE terms based on the 2.5 km-EC simulation with 1-moment cloud microphysics. The figure is for 23 September 2016, 12 UTC. The cyclone position as given by the minimum mean sea level pressure is marked by the blue cross.

The equation decomposes the surface pressure tendency (Dp) into stratospheric changes that manifest in the geopotential at the upper boundary of the vertical intergral (chosen here as 100 hPa; D$\phi$), changes in the tropospheric virtual temperature





(ITT), and changes in column mass due to evaporation and precipitation (E-P). Because the E-P term is very small, we do not

calculate it explicity but absorb it into the residual term RES$_{PTE}$.

Tropospheric heating leads to a drop in surface pressure. We are mostly interested in tropospheric heating and therefore decompose the ITT term further,

$$
ITT = + \rho_{\text{sfc}} R_{\text{d}} \int_{\text{sfc}}^{100\,\text{hPa}} -\boldsymbol{v} \cdot \boldsymbol{\nabla}_{\text{p}} T_{\text{v}} \, \text{d}(\ln p) \qquad (TADV)
$$

$$
+ \rho_{\text{sfc}} R_{\text{d}} \int_{\text{sfc}}^{100\,\text{hPa}} \left( \frac{R_{\text{d}} T_{\text{v}}}{c_{\text{p}} p} - \frac{\partial T_{\text{v}}}{\partial p} \right) \omega \, \text{d}(\ln p) \qquad (VMT)
$$

$$
+ \rho_{\text{sfc}} R_{\text{d}} \int_{\text{sfc}}^{100\,\text{hPa}} \frac{T_{\text{v}} Q}{c_{\text{p}} T} \, \text{d}(\ln p) \qquad (DIAB)
$$

$$
+ \text{RES}_{\text{ITT}},
$$

where $T$ is the temperature, $\boldsymbol{v}$ and $\omega$ are the horizontal and vertical components of wind, $c_p$ is the specific heat capacity at

constant pressure and $Q$ is the diabatic heating rate. On the right-hand side of the equation, the first two terms represent the impact of horizontal temperature advection (TADV) and vertical motions (VMT) on the ITT. DIAB represents the influence of diabatic heating. The term RES$_{\text{ITT}}$ represent errors caused by temporal and spatial discretizations, similar to RES$_{\text{PTE}}$. Following Fink et al., 2012 and Pohle, 2010 we measure the impact of diabatic heating as the residuum of ITT and the horizontal and advective terms,

$$
\text{DIAB}_{\text{res}} = \text{DIAB} + \text{RES}_{\text{ITT}} = \text{ITT} - (\text{TADV} + \text{VMT})
$$

We calculate the PTE and its decomposition using hourly model output that is interpolated from the 75 model levels onto pressure levels with a vertical spacing of 10 hPa. For illustration, Fig. 10 shows maps of the PTE terms for the 2.5 km-EC simulation at 23 Sep 2016, 12 UTC. The overall pattern is similar across the model setups. Near the cyclone centre, a dipole pattern of negative and positive Dp values is visible, which mainly results from the ITT term. The ITT term itself is characterized by large and opposing impacts from horizontal advection (TADV) and vertical motion (VMT), as well as negative surface pressure

tendencies from diabatic heating in the region of the WCB. However, near the cyclone centre diabatic heating has a relatively small and in fact positive impact.

We now assess how the PTE terms contribute to the deepening of cyclone Vladiana. Fig. 11 depicts the time series of the PTE terms averaged over a $3° \times 3°$ latitude-longitude box centred around the cyclone location. Overall, the evolution of the cyclone central pressure is most strongly affected by tropospheric heating (i.e., the ITT term; top panel), which itself is dominated by

horizontal temperature advection (TADV; middle panel). Diabatic heating plays a smaller role and does not contribute to the cyclone deepening but works against it.

The dominant role of horizontal advection and the minor impact of diabatic heating is robust across model setups. To show this, Fig. 12 depicts the PTE terms averaged over the main deepening period of the cyclone. The somewhat less intense cyclone

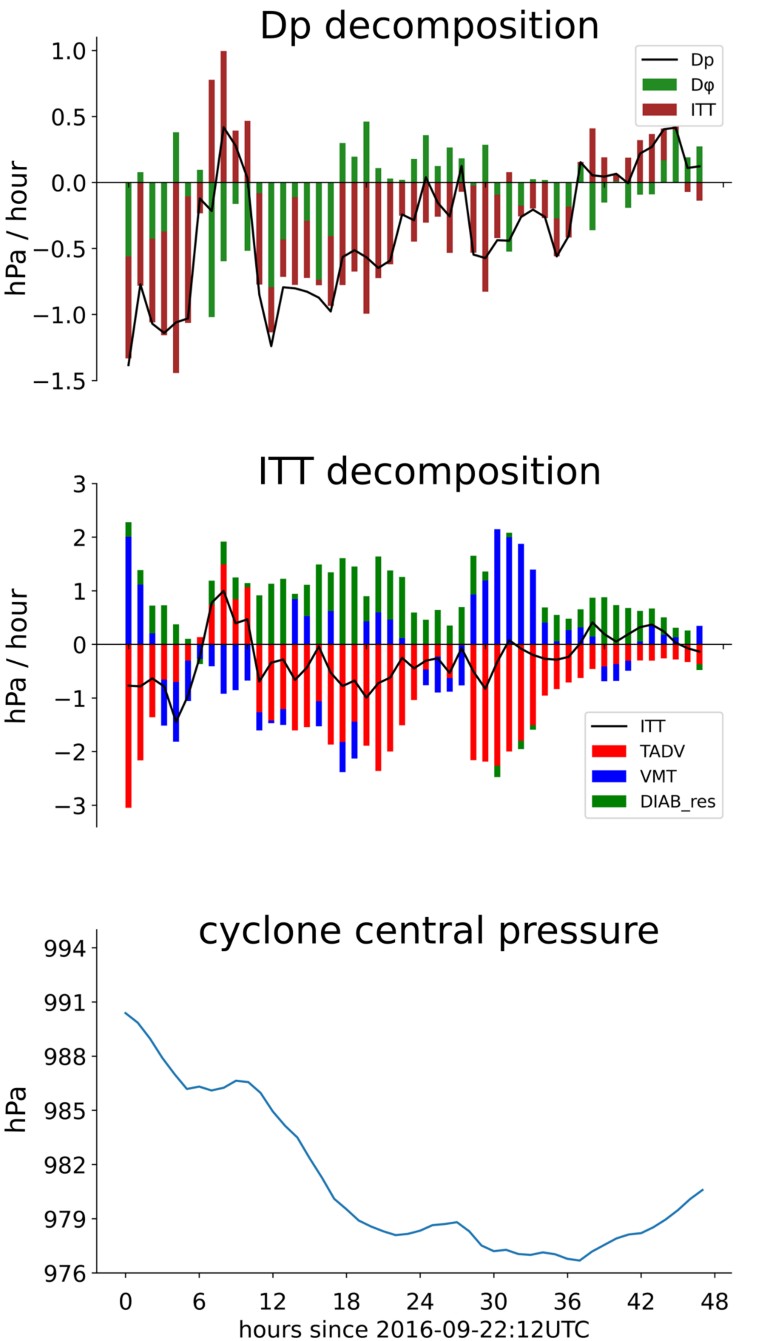

**Figure 11.** Time series of the PTE analysis during the cyclone development since 22 September 2016 12 UTC for the 2.5 km-EC simulation with 1-moment cloud microphysics. The PTE terms are averaged over a $3° \times 3°$ latitude-longitude box centred around the cyclone location. Top: surface pressure tendency and its decomposition. Middle: Decomposition of the ITT term. Bottom: central pressure of the cyclone.



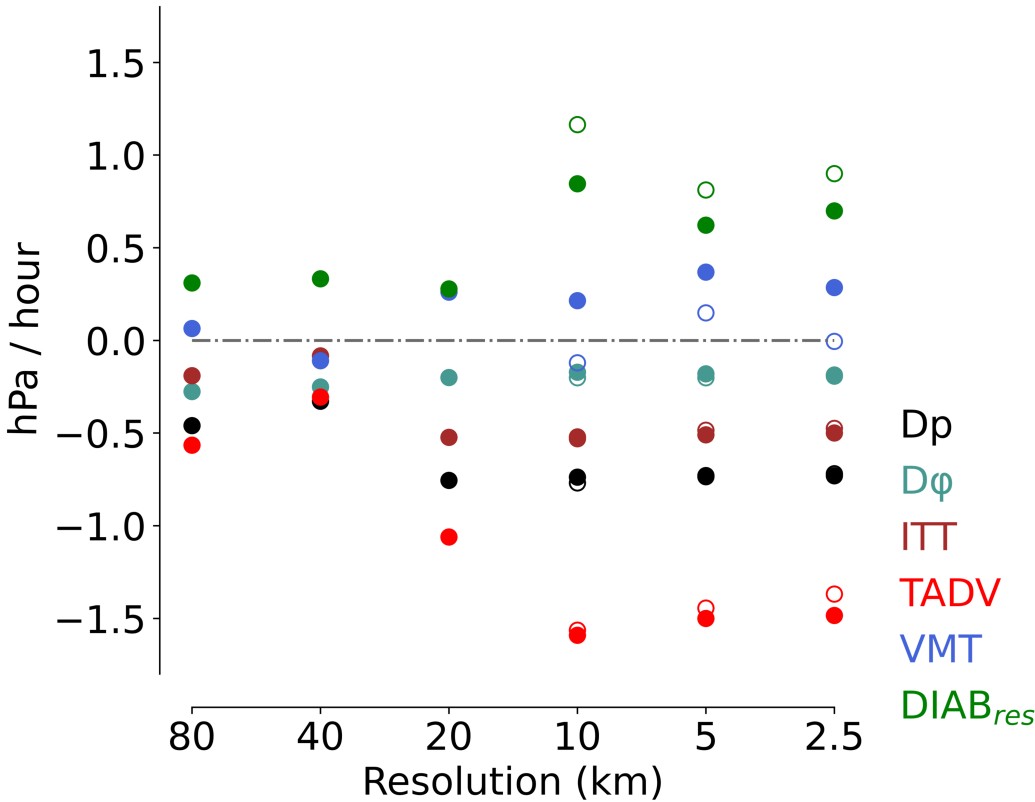

**Figure 12.** PTE terms as a function of horizontal resolution for simulations with 1-moment cloud microphysics. The terms are averaged over hours 10 to 21 since 22 September 2016 12 UTC. This period is selected to focus on the continuous deepening of the cyclone. Filled and empty markers represent simulations with parametrized and explicit convection, respectively.

for a resolution of 80 and 40 km arises from a smaller contribution of temperature advection. In contrast, diabatic heating for all resolutions works against the cyclone deepening.

In summary, the PTE analysis shows that the deepening of cyclone Vladiana is not due to diabatic processes. This explains why the systematic enhancement of diabatic processes in the warm conveyor belt that we have documented in Sect. 3 is not reflected in the synoptic evolution of the cyclone in terms of its central pressure.

## 5 Conclusions

We have characterized how the simulation of a warm conveyor belt (WCB) of a midlatitude cyclone is affected by model resolution and the treatment of convection and microphysics. Our study is motivated by the development of global and regional storm-resolving models that aim to represent the atmosphere with a horizontal resolution of a few kilometer and without a deep convection scheme. It is further motivated by previous results that midlatitude cyclones intensify as resolution is increased.





We have analyzed a set of simulations with the ICON model in limited-area setup over the North Atlantic and with the atmospheric physics package developed for numerical weather prediction. The simulations were run for horizontal resolutions ranging from 80 to 2.5 km, with and without a convection scheme, and with 1-moment and 2-moment cloud microphysics. The simulations were a case study of the North Atlantic cyclone Vladiana, which occured in September 2016 and exhibited a well-developed WCB. Our analysis has used offline trajectories and the surface pressure tendency equation (PTE).

Based on the analysis we answer three research questions given in the introduction as follows:

1. How do horizontal resolution, the treatment of convection and the treatment of cloud microphysics affect the WCB ssociated with cyclone Vladiana?

   As resolution is increased, the number of WCB trajectories increases by up to a factor of 10. When convection is represented explicitly, the number of WCB trajectories increases further. For the highest resolutions of 10, 5 and 2.5 km and with explicit convection, the number of WCB trajectories does not depend on resolution, signaling convergence. We find analogous impacts of resolution and the treatment of convection on the WCB ascent and vorticity, which both strengthen for increased resolution and explicit convection. Cloud microphysics have a minor impact.

2. How sensitive is diabatic heating within the WCB to these modeling choices?

   As resolution is increased, diabatic heating systematically increases. The increase arises from stronger diabatic heating by cloud microphysics, and is consistent with stronger WCB ascent and hence stronger latent heating. For parametrized convection, each increase in resolution leads to an increase in diabatic heating. When convection is treated explicitly in the 10, 5 and 2.5 km simulations, diabatic heating is largely insensitive to resolution. The impact of the treatment of cloud microphysics is again minor.

3. Do the sensitivities of the WCB diabatic processes affect the deepening of cyclone Vladiana?

   We do not find a clear and systematic impact of model resolution and the treatment of convection on the evolution of the central pressure of cyclone Vladiana. This is in contrast to the above sensitivities of the diabatic heating. The difference is explained by the PTE analysis, which shows that the deepening of cyclone Vladiana is driven by temperature advection and not by diabatic processes.

A limitation of our study is that we have not compared the simulations to observational data and so cannot quantify the added value of increasing resolution and disabling the convection scheme. Nevertheless, a few points can be made. For the coarse resolutions of 80 and 40 km, the WCB is much weaker and much less pronounced compared to the other resolutions. For the coarsest resolution of 80 km, only 3 of the 4 trajectory subclasses are simulated, hinting at a possible systematic shortcoming of low-resolution models that might impact the waveguide and thus the downstream flow evolution (Oertel et al., 2020).

Our results further indicate that when the convection scheme is switched off, a resolution of 5 km or maybe even 10 km is sufficient. The results from these resolutions are in close agreement with the results from the 2.5 km simulation with explicit convection in terms of the WCB characteristics, the WCB diabatic heating and the deepening of the cyclone. This finding is



broadly consistent with Vergara-Temprado et al., 2020, who reported that at a resolution of 20 km and finer the representation of deep convection plays a larger role than a further increase in resolution. The finding is also consistent with Jung et al., 2006, Champion et al., 2011 and Jung et al., 2012 that a resolution of 20 km is sufficient to capture the synoptic evolution of midlatitude cyclones.

For future work we would find it interesting to investigate simulations in a Transpose-AMIP framework (Williams et al., 2013) in which climate models are used to predict weather over the course of around 10 days. Because T-AMIP simulations start from a known state of the atmosphere, the impact of modelling choices on midlatitude cyclones could be studied across a large number of cyclones and the results could be evaluated by means of reanalysis and observational data.

*Data availability.* The data that support the findings of this study are openly available. The analysis scripts are provided in the Gitlab
repository https://gitlab.phaidra.org/climate/choudhary-vladiana-wcd-2022 hosted by University of Vienna. The WCB trajectory output from LAGRANTO and the other processed data from simulations used in the work is published at Zenodo with doi 10.5281/zenodo.5921126 (https://doi.org/10.5281/zenodo.5921126). The Zenodo data set also includes a copy of the analysis scripts.

*Author contributions.* The ICON simulations were carried out by A.V. A.C. and A.V. designed the study. A.C. did the analysis with inputs from A.V. Both the authors discussed, interpreted the results and wrote the paper.

*Competing interests.* The authors declare that they have no conflict of interest.

*Acknowledgements.* A.C. and A.V. acknowledge financial support by the German Ministry of Education and Research (BMBF) and FONA: Research for Sustainable Development (www.fona.de) under grant agreement 01LK1509A. The ICON simulations were carried out by A.V. at the Mistral High Performance Computing system of the German Climate Computing Center (DKRZ) in Hamburg, Germany. The primary data of the ICON simulations (run scripts, namelists, scripts for lateral boundary data) are published at KITopen of Karlsruhe Institute
of Technology, https://doi.org/10.5445/IR/1000123695. Note that the KITopen dataset includes all simulations of Senf et al., 2020, from which a subset is analyzed here. This work used resources of the Deutsches Klimarechenzentrum (DKRZ) granted by its Scientific Steering Committee (WLA) under project ID bb1018. This work contributes to the WCRP's Grand Challenge on Clouds, Circulation, and Climate Sensitivity and the BMBF-funded project "HD(CP)$^2$: High Definition Clouds and Precipitation for Advancing Climate Prediction". We are very thankful to Georgios Papavasileiou of National Observatory of Athens, Greece for his help with the PTE analysis. We also thank Annika
Oertel of IMK-TRO, KIT, Germany for feedback and discussions.



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
