# Peer review of "Impact of grid spacing, convective parameterization and cloud microphysics in ICON simulations of a warm conveyor belt"

_Weather and Climate Dynamics, 2022_

## Referee Comment (RC1)

**Review of «ICON simulations of cloud diabaitc processes in the warm conveyor belt of North Atlantic cyclone Vladiana" by A. Choudhary and A. Voigt**

In this paper, the representation of the warm conveyor belt (WCB) which is associated with the extratropical cyclone Vladiana in ICON simulations is discussed. The authors compare in detail the characteristics of the WCB in 18 simulations with 6 different horizontal resolutions, raning from 80km to 2.5 km, parameterized and explicit convection, as well as 1 and 2-moment cloud microphysical parameterizations. The presented analysis is a valuable contribution to the existing literature as it is not well understood how the representation of WCBs and their impact on cyclone evolution, precipitation, the upper tropospheric flow structure and the downstream flow evolution will change with increasing resolution and the explicit treatment of convection in future weather and climate models.
Although the authors do not discuss all of these implications but focus on cyclone intensification, I would recommend to accept this publication with minor revisions.

Minor comments:
- For all model resolutions you use offline trajectories which are calculated based on 1-hourly output fields. While I see the reason for that I think it would be good if you included some more critical discussion about this fact. In line 221 you say that the slower ascent is caused by the fact that you use offline instead of online trajectories (in comparison to Oertel et al., 2019). However I'm not so convinced that this is necessarily the case. By using the instantaneous value of the vertical velocity every hour in your offline trajectories it could also be that you overestimate the ascent. In case the trajectory at this point in time sees very high convective updrafts which might be relatively short lived, you assume that this high velocity lasts for one hour. This might not be so important when going to coarser resolutions but for the high resolution simulations I think it could happen. Or maybe this effects cancels in the mean over all trajectories as some of them also assume "unrealistic small updrafts speeds" over one hour? Could you add some more thoughts on this to your text?
- I think it is interesting to see that the number of trajectories in the trajectory 2 cluster strongly increases whereas this increase is less pronounced for the other ones. I wonder what the reason for this is? Do you know where these trajectories originate in comparison to the others? Are they especially convective? Are there special mesoscale substructures e.g. in moisture and or theta values in the higher resolution simulations where these trajectories originate? Or what is the evolution of different parameters like, p, theta, q, ….along this cluster compared to the others? Maybe this is out of the scope of your study but maybe you have thoughts on this which you could include?
- Lines 115 ff: The synoptic description could be a little bit more precise, e.g. in line 118 you say that there are low PV values north of the British Isles, however they are low over the whole Island including Ireland….and in Line 118: you state that the cyclone is decaying in the next 24h, however the minimum presses in Fig. 1c is still at 985 hPa → please adapt the description of the synoptic evolution.
- End of chapter 4: You show that the deepening of Vladiana is not due to diabaitc processes and that in this case the WCB does not contribute to the cyclone intensification. However is it possible that the WCB contributes e.g. to the formation or strengthening of PV anomalies at the cyclonens warm and cold front and that there is a

difference between the different resolutions? Could a better representation of fronts and the diabatic processes in there in the higher resolution datasets lead to changed mesoscale weather phenomena, like enhanced precipitation or different precipitation patterns, changed PV anomalies associated with also differences in the wind speed?

Style comments:

- Line 91: ..with analysis and forecast data from the ECMWF……why do you use forecast data here and not only analysis data ?
- E.g. line 105: make sure that the date formats are the same everywhere
- Line 158 end: "… to be be…" → remove one be
- Figure 2 caption: trajectory 3 (red) , should be "yellow" ?
- Figure 5 caption, 2. Line: …,with given by the right x-axis → right y-axis.
- Figure 5: Panels a and b should be horizontally aligned, or maybe even remove panel b as not much is changing and only mention it in the text
- Figure 6: Aling as 2X2 figure, not all 4 in a column
- Line 262: Section title: "4 Synoptic evolution of cyclone Vladiana ….." → "4 Pressure evolution of …."
- Figure 8: align panels horizontally and vertically.
- Line 293: intergral –> integral
- In the reference list there are still many typos (e.g. capital letters, "" signs…..), please correct

---

## Referee Comment (RC2)

Review

**ICON simulations of cloud diabatic processes in the warm conveyor belt of North Atlantic cyclone Vladiana**

This manuscript contains a nice, detailed, and insightful analysis of 18 ICON simulations of cyclone Vladiana. The authors focus on the dynamics of the warm conveyor belt and on its relationship with model horizontal resolution, convection settings and microphysics scheme. This is a well-written piece of research that could surely be a welcome contribution to the field. However, I have some reservations on the application of the PTE method (see the two relevant comments at the end of this document) and, as such, I don't think the third research question (see Conclusions) has been fully answered as of now. Apart from this, all other comments are minor and, in some cases, I'm just requesting to expand some discussions and clarify some statements. Overall, I think this paper would be worth of publication once the comments are addressed.

**Comments:**

**General:**

- I find the title slightly vague and I don't think it does justice to the novelty of your analysis. I appreciate that for space reasons it won't be possible to explicitly mention model resolution, convective parameterisation and microphysics scheme, but I would suggest you try highlighting the analysis performed on these ICON simulations.

- I would consider replacing "higher/highest" and "lower/lowest" resolution with "smoother/smoothest" and "coarser/coarsest", respectively. Given that grid spacing decreases with finer ("higher") resolution, those two terms might be misleading (the same applies to all the "increasing/decreasing resolution" instances).

-DOIs seem to be missing from most of the references included in this manuscript. It would be helpful for reviewers & readers to add them, so the referenced research can be accessed more quickly.

**Abstract:**

- Lines 7-8: "the resolution" here could be replaced by "that" in both instances, making the sentence more concise.

- Lines 13: "With increasing resolution". I would specify exactly what are the two resolutions that show a 3-time increase in ascent velocity. Also, if there's space, I would include a brief statement on the saturation of those effects when moving to resolutions finer than 10 km.

- Lines 17-18: "on the one hand" and "on the other hand" are not really needed here.

Introduction:

- General: There is a sort of jump between the first two paragraphs, describing the WCB, and the following one, starting with "Despite decades of model development…". I would suggest improving the link between the different topics presented and/or to try change the order of paragraphs and see if the readability improves.

- Lines 23-24: "Most of the diabatic processes occur within coherent streams of ascending air known as warm conveyor belts". This is a bold claim and I don't think the papers referenced justify it. I understand the reasoning behind it, but I would still like it to be rephrased (or properly justified) as substantial diabatic processes can be found in many other parts of the cyclone, even outside the warm sector.

- Lines 44-45: It might be better saying ".. in which horizontal grid spacing is reduced to a few kilometers.." (or something similar, should you prefer taking into account spectral models).

Method:

- Lines 89-90: Could you give some motivation as to why it is ok that the model follows a " tropical Atlantic setup"?

- Line 91: Consider moving the domain description here (or motivate why it should stay at lines 106-107).

- Line 94: Did you consider running half-explicit convection (i.e., with only shallow or deep parameterisations disabled)? I'm not asking you to run anymore simulations, I was just wondering if you thought such as setup could be useful (possibly in future works?)

- Line 105: Could you briefly specify here how what the 18 simulations are? (I don't think a table is needed, a sentence in the text should be enough).

- Figure 1: I would replace the PV colour scale with a more intuitive (and colour-blind friendly) one. I understand the emphasis on the 2 PVU value, but I think it could just by highlighted with a red contour line, while using a more logical colour scale for the shading. Also, I would replace 'EQ' with '0W' on the x axis.

- Line 135-137: "from every grid point": this would suggest that simulations with finer resolutions have more trajectory starting points, whereas line 137 indicates that this is not the case as "the seeding points are based on the 20 km simulation. For all resolutions….". I assume the latter is true but a clarification is needed, for example moving this second quote up to line 135 (and, if my assumption is wrong, "number" needs to be replaced with "proportion" at line 9).

- Line 147: "As a result, …" could you clarify why this point about slantwise convection is a consequence of using off-line rather than on-line trajectories?

- 2.4 Diabatic Heating: Are these tendencies computed on-line by the model?

WCB trajectories:

- Lines 177-178: Is it possible to be more quantitative here? (giving more context to the following split into subsets, and Figure 4)

- Line 182: I would say "more" than 10 times, looking particularly at Figure 3.

- Lines 185-187: Again, could you make this statement more quantitative?

- Figure 4: How are the boundaries of the rectangular domains defined? Is there any motivation to not just stop at an "anticyclonic vs cyclonic" decomposition?

- Figure 5: I would consider changing this figure to a 4-panel one, to separate total and subclasses counts. other solutions are welcome, as long as they make it clearer to interpret (I find having all and subclasses together, and on two different scales, slightly confusing).

- Figure 6: I suggest making this a 2x2-panel figure (a similar consideration applies to Fig 8, which could become a 3x2 or 2x3 one). I think that using colours would help differentiating between the various points. Also, I suppose that "mean ascent height" indicates the mean pressure difference between beginning and end of the ascent, rather than the pressure value half-way through the ascent. I think a clearer name for this quantity might be needed (if instead it is the latter, the y-axis should be reversed).

- Lines 216-217: Based on this result, can we say that in coarser simulations we see fewer WCB trajectories because generally ascent is less vigorous? Or are there other factors at play?

- Line 221: Can you elaborate on the reasons behind this statement? (slower ascent because trajectories are off-line)

- Line 234: I would add a rough timing of the two main periods.

- Fig 7: Given that $\Delta h$ is approximately proportional to $-\Delta(\log P)$, have you considered using log-scale in all panels (in Fig 7 elsewhere in the manuscript) where pressure is on the x or y axis?

- Line 239: I completely agree with you on the expected PV vertical profile, but this might not be familiar to all readers. Could you add a reference?

- Lines 249-250: I'm not sure I agree with this sentence. Even if the output of a 2.5km simulation is regridded to 40km, that simulation was run with the model being able to resolve processes well below 40km (although I appreciate that coarsening the output will smooth out the patterns).

- Figure 9: adding a line for the 2.5km simulation regridded at 40km and/or 80km would help the argument that horizontal resolution does not influence cyclone deepening.

- Section 4.1: Would it be possible to compute PTE in a storm-relative frame of reference or, even better, to use a fully Lagrangian perspective and trace the terms onto the trajectories computed? I'm saying this as I'm thinking at ways to reduce the impact of advection and the related cancellations between terms.

- Line 304: I find slightly surprising that the diabatic term is not calculated explicitly , but instead grouped with the residuals. Without knowing how big residuals are, I don't think it's possible making conclusions on the behaviour of the diabatic term.

---

## Author Comment (AC1)

**ICON simulations of cloud diabatic processes in the warm conveyor belt of North Atlantic cyclone Vladiana - Response to Reviewers**

**Anubhav Choudhary and Aiko Voigt**

We thank the reviewers for their encouraging evaluations and suggestions to improve our manuscript. Below, we respond in detail to each of the reviewers' comments. We are confident that we can address all of their comments and describe how we plan to revise our manuscript. The reviewers' comments are in bold, our answers are in normal font.

In the revised paper, we will also include references to recent work on WCBs and extratropical cyclones. This includes, among others, the work of Blanchard et al. (2020); Flack et al. (2021) and Mazoyer et al. (2021).

**Reviewer 1**

**In this paper, the representation of the warm conveyor belt (WCB) which is associated with the extratropical cyclone Vladiana in ICON simulations is discussed. The authors compare in detail the characteristics of the WCB in 18 simulations with 6 different horizontal resolutions, ranging from 80km to 2.5 km, parameterized and explicit convection, as well as 1 and 2-moment cloud microphysical parameterizations. The presented analysis is a valuable contribution to the existing literature as it is not well understood how the representation of WCBs and their impact on cyclone evolution, precipitation, the upper tropospheric flow structure and the downstream flow evolution will change with increasing resolution and the explicit treatment of convection in future weather and climate models. Although the authors do not discuss all of these implications but focus on cyclone intensification, I would recommend to accept this publication with minor revisions.**

Thank you for your interest in our work!

**Minor comments:**

**For all model resolutions you use offline trajectories which are calculated based on 1-hourly output fields. While I see the reason for that I think it would be good if you included some more critical discussion about this fact. In line 221 you say that the slower ascent is caused by the fact that you use offline instead of online trajectories (in comparison to Oertel et al., 2019). However I'm not so convinced that this is necessarily the case. By using the instantaneous value of the vertical velocity every hour in your offline trajectories it could also be that you overestimate the ascent. In case the trajectory at this point in time sees very high convective updrafts which might be relatively short lived, you assume that this high velocity lasts for one hour. This might not be so important when going to coarser resolutions but for the high resolution simulations I think it could happen. Or maybe this effects cancels in the mean over all trajectories as some of them also assume "unrealistic small updrafts speeds" over one hour? Could you add some more thoughts on this to your text?**

As noted by the reviewer, the use of offline trajectories is a pragmatic choice for the simple reason that ICON-NWP v2.1 does not include online trajectories (these have only been only implemented into a later version of ICON). We agree that the effect described by the reviewer is possible, yet Fig. S2 of the supplemental material of our original submission shows that the effect plays a minor role. Even at the finest grid spacing and with explicit convection, we find no trajectories with hourly ascent rates above 600 hPa, and

we find only very few trajectories with rates above 400 hPa. This is different from Fig. 5c of Oertel et al. (2019), for which such ascent rates occur rather frequently. As a result, the diagnosed mean ascent is slower for offline trajectories compared to online trajectories.

We will revise the manuscript accordingly to clarify this point.

**I think it is interesting to see that the number of trajectories in the trajectory 2 cluster strongly increases whereas this increase is less pronounced for the other ones. I wonder what the reason for this is? Do you know where these trajectories originate in comparison to the others? Are they especially convective? Are there special mesoscale substructures e.g. in moisture and or theta values in the higher resolution simulations where these trajectories originate? Or what is the evolution of different parameters like, p, theta, q, ... along this cluster compared to the others? Maybe this is out of the scope of your study but maybe you have thoughts on this which you could include?**

We agree that this is an interesting aspect of the trajectory clusters - thank you for bringing it up. The ascent occurs in two main phases. A first bundle of trajectories starts the ascent at around 2016-09-23:00UTC; a second bundle starts the ascent at around 2019-09-23:18UTC. The increase in the number of trajectories in class 2 as the grid spacing is decreased results from an increasing number of trajectories in the second bundle. We will include this information in the revised manuscript.

**Lines 115 ff: The synoptic description could be a little bit more precise, e.g. in line 118 you say that there are low PV values north of the British Isles, however they are low over the whole Island including Ireland... .and in Line 118: you state that the cyclone is decaying in the next 24h, however the minimum pressure in Fig. 1c is still at 985 hPa. Please adapt the description of the synoptic evolution.**

We agree and will adapt the description in the revised manuscript.

**End of chapter 4: You show that the deepening of Vladiana is not due to diabatic processes and that in this case the WCB does not contribute to the cyclone intensification. However is it possible that the WCB contributes e.g. to the formation or strengthening of PV anomalies at the cyclone's warm and cold front and that there is a difference between the different resolutions? Could a better representation of fronts and the diabatic processes in there in the higher resolution datasets lead to changed mesoscale weather phenomena, like enhanced precipitation or different precipitation patterns, changed PV anomalies associated with also differences in the wind speed?**

We agree with the reviewer that these phenomena would be interesting to study. Our interest in the paper is to test whether there is a link between the large-scale behavior of the cyclone (i.e., its deepening) and the simulation of WCB diabatic processes as a function of grid spacing and model physics, and so we prefer not to touch upon these questions in our analysis. However, we will revise the paper to mention possible WCB impacts beyond the cyclone core pressure, and to this end plan to specifically include the recent work of Oertel et al. (2019, 2020); Flack et al. (2021); Blanchard et al. (2020, 2021); Rivière et al. (2021) and Wimmer et al. (2021).

In Section 4, we also plan to interpret the weak WCB impact on the deepening of Vladiana using Binder et al. (2016), whose framework suggests that Vladiana is (in their terminology) a C2 cyclone with a strong WCB, but with a WCB that is far away from the cyclone center and ascents mainly along the cold front.

**Style comments:**

**Line 91: ..with analysis and forecast data from the ECMWF ... why do you use forecast data here and not only analysis data?**

Note that we use data from the ECMWF operational model IFS at the highest available resolution (around 9 km) (not ERA5). The lateral boundary data is updated every 3 hours. Analysis data is available at 0 and 12 UTC, in between 3-, 6-, and 9-hr IFS forecast data are used. This is described in Senf et al. (2020). In the revised manuscript we will clarify the initial and boundary data, and will replace the reference to Klocke et al. (2017) by Senf et al. (2020).

**E.g. line 105: make sure that the date formats are the same everywhere**

Thank you, will do.

**Line 158 end: "...to be be ..."; remove one be**

Thanks.

**Figure 2 caption: trajectory 3 (red), should be "yellow"?**

Yes, indeed. We will correct this.

**Figure 5 caption, 2. Line: ..., with given by the right x-axis → right y-axis**

Thanks.

**Figure 5: Panels a and b should be horizontally aligned, or maybe even remove panel b as not much is changing and only mention it in the text**.

We will realign the figure. Yet, we prefer to keep the panel b with the simulations using the 2-moment microphysics to explicitly show that microphysics play a minor role.

**Figure 6: Align as 2X2 figure, not all 4 in a column**

Good suggestion, thank you.

**Line 262: Section title: "4 Synoptic evolution of cyclone Vladiana ..." → "4 Pressure evolution of ..."**

Thanks, we will use "pressure evolution" in the revised manuscript to make it clear that we only consider the cyclone core pressure in this section.

**Figure 8: align panels horizontally and vertically**

Yes, we will realign the panels.

**Line 293: integral → integral**

Thanks, this will be corrected.

**In the reference list there are still many typos (e.g. capital letters, " " signs, ...), please correct.**

Thank you, and please accept our apologies for not having checked the reference list carefully. We will correct this in the revised manuscript.

**Reviewer 2**

**This manuscript contains a nice, detailed, and insightful analysis of 18 ICON simulations of cyclone Vladiana. The authors focus on the dynamics of the warm conveyor belt and on its relationship with model horizontal resolution, convection settings and microphysics scheme. This is a well-written piece of research that could surely be a welcome contribution to the field. However, I have some reservations on the application of the PTE method (see the two relevant comments at the end of this document) and, as such, I don't think the third research question (see Conclusions) has been fully answered as of now. Apart from this, all other comments are minor and, in some cases, I'm just requesting to expand some discussions and clarify some statements. Overall, I think this paper would be worth of publication once the comments are addressed.**

Thank you for your interest in our work!

**General comments:**

**I find the title slightly vague and I don't think it does justice to the novelty of your analysis. I appreciate that for space reasons it won't be possible to explicitly mention model resolution, convective parametrization and microphysics scheme, but I would suggest you try highlighting the analysis performed on these ICON simulations.**

Thank you, we will change the title to "Impact of grid spacing and model physics in ICON simulations of a warm conveyor belt". The new title highlights the analysis approach. The words "cloud diabatic" and "North Atlantic cyclone Vladiana" will be cut as they are not essential, allowing for an in fact shorter title.

**I would consider replacing "higher/highest" and "lower/lowest" resolution with "smoother/smoothest" and "coarser/coarsest", respectively. Given that grid spacing decreases with finer ("higher") resolution, those two terms might be misleading (the same applies to all the "increasing/decreasing resolution" instances).**

We agree that this can be confusing. We will replace "resolution" by "grid spacing" throughout the manuscript and will adapt the sentences accordingly.

**DOIs seem to be missing from most of the references included in this manuscript. It would be helpful for reviewers & readers to add them, so the referenced research can be accessed more quickly.**

We will include dois for all references.

**Comments regarding the abstract:**

**Lines 7-8: "the resolution" here could be replaced by "that" in both instances, making the sentence more concise.**

Agreed.

**Lines 13: "With increasing resolution". I would specify exactly what are the two resolutions that show a 3-time increase in ascent velocity. Also, if there's space, I would include a brief statement on the saturation of those effects when moving to resolutions finer than 10 km.**

Agreed.

**Lines 17-18: "on the one hand" and "on the other hand" are not really needed here.**

We will remove "on the one hand" and "in terms of its central pressure on the other hand".

**Introduction:**

**There is a sort of jump between the first two paragraphs, describing the WCB, and the following one, starting with "Despite decades of model development...". I would suggest improving the link between the different topics presented and/or to try change the order of paragraphs and see if the readability improves.**

Thank you, we will try to rephrase these paragraphs to improve the flow.

**Lines 23-24: "Most of the diabatic processes occur within coherent streams of ascending air known as warm conveyor belts". This is a bold claim and I don't think the papers referenced justify it. I understand the reasoning behind it, but I would still like it to be rephrased (or properly justified) as substantial diabatic processes can be found in many other parts of the cyclone, even outside the warm sector.**

Agreed – we will rephrase. In fact, we did not mean to imply that diabatic processes are limited to the WCB.

**Lines 44-45: It might be better saying "... in which horizontal grid spacing is reduced to a few kilometers ..." (or something similar, should you prefer taking into account spectral models).**

We agree and will rephrase as suggested.

**Method:**

**Lines 89-90: Could you give some motivation as to why it is ok that the model follows a "tropical Atlantic setup"?**

Sorry for the confusing formulation. What we meant is that our simulations use essentially the same model setup (apart from the domain location) as the (sub)tropical North Atlantic simulations of Klocke et al. (2017). However, the formulation in the original manuscript was vague and misleading. We will remove the phrase regarding the "tropical Atlantic setup" and will instead refer to Senf et al. (2020). Senf et al. (2020) analyzed the same set of North-Atlantic simulations that we study in our paper, and provided a detailed description of the model setup.

**Line 91: Consider moving the domain description here (or motivate why it should stay at lines 106-107).**

Thank you, we will consider this in the revision.

**Line 94: Did you consider running half-explicit convection (i.e., with only shallow or deep parameterizations disabled)? I'm not asking you to run anymore simulations, I was just wondering if you thought such as setup could be useful (possibly in future works?)**

One of us (AV) has performed and analyzed simulations with active shallow but disabled deep convective parameterization. These simulations are included in Senf et al. (2020), who evaluated them in terms of North Atlantic clouds and top-of-atmosphere radiative effects. The simulations are not included here for the pragmatic reason that they were run only after the technical aspects of the WCB trajectory analysis was already finished.

**Line 105: Could you briefly specify here how what the 18 simulations are? (I don't think a table is needed, a sentence in the text should be enough).**

Yes, we will such a sentence in the revised manuscript.

**Figure 1: I would replace the PV colour scale with a more intuitive (and colour-blind friendly) one. I understand the emphasis on the 2 PVU value, but I think it could just by highlighted with a red contour line, while using a more logical colour scale for the shading. Also, I would replace "EQ" with "0W" on the x axis.**

We wish to keep the PV color scale in its current form as it mimics the color scale of Fig. 2 d-f of Oertel et al. (2020). We will make this choice clear in the caption. And yes, on the x-axis "EQ" should of course be "0W"!

**Line 135-137: "from every grid point": this would suggest that simulations with finer resolutions have more trajectory starting points, whereas line 137 indicates that this is not the case as "the seeding points are based on the 20 km simulation. For all resolutions...". I assume the latter is true but a clarification is needed, for example moving this second quote up to line 135 (and, if my assumption is wrong, "number" needs to be replaced with "proportion" at line 9).**

Thank you, the formulation "from every grid point" was indeed confusing. We will clarify this in the revised manuscript. In particular, we will mention that the number of seeded trajectories is the same for all simulations, and in fact is 395,825.

**Line 147: "As a result, ..." could you clarify why this point about slantwise convection is a consequence of using off-line rather than on-line trajectories?**

The slantwise character follows from two points. First, in the coarse-grid simulations the convection scheme is active and the grid spacing is too large to allow for convective updrafts by the resolved flow. Second, the fine-grid simulations allow for convective updrafts – however, the updrafts are largely missed as they are short-lived and the wind-field is sampled only hourly. Fig. S2 in the supplemental material illustrates the largely slantwise character of the ascent: the region of strong updrafts is unpopulated, signaling that the ascent is slantwise. Yet, we realize now that we did not refer to Fig. S2 in the original submission. In the revised manuscript we will include the above discussion and a reference to Fig. S2.

**2.4 Diabatic Heating: Are these tendencies computed on-line by the model?**

Yes, the diabatic tendencies are diagnosed online.

**WCB trajectories:**

**Lines 177-178: Is it possible to be more quantitative here? (giving more context to the following split into subsets, and Figure 4)**

Yes, good idea. We will include the trajectory numbers and link the text to Fig. 4.

**Line 182: I would say "more" than 10 times, looking particularly at Figure 3.**

Agreed, we will rephrase to "more than 10 times".

**Lines 185-187: Again, could you make this statement more quantitative?**

We will make the statement more quantitative. In essence, the number of trajectories increases by 30% between the 10 km and 2.5 km grid spacing for parameterized convection, but does not change systematically with grid spacing for explicit convection.

**Figure 4: How are the boundaries of the rectangular domains defined? Is there any motivation to not just stop at an "anticyclonic vs cyclonic" decomposition?**

The boundaries were chosen based on two reasons. First, we wanted to separate cyclonic and anticyclonic trajectories. Then, given the high number of anticyclonic trajectories and their clear differences in final location, we further separated anticyclonic trajectories into 3 classes. The distinct behavior of class 4, which turn southward very quickly, was a particular motivation for this additional separation.

**Figure 5: I would consider changing this figure to a 4-panel one, to separate total and subclasses counts. Other solutions are welcome, as long as they make it clearer to interpret (I find having all and subclasses together, and on two different scales, slightly confusing).**

Thank you. However, we prefer to keep the figure with all trajectories and the subclasses in the same panel. This makes it easier to compare how the grid-spacing dependence of the number of all trajectories relates to the subclasses.

**Figure 6: I suggest making this a 2x2-panel figure (a similar consideration applies to Fig 8, which could become a 3x2 or 2x3 one). I think that using colours would help differentiating between the various points. Also, I suppose that "mean ascent height" indicates the mean pressure difference between beginning and end of the ascent, rather than the pressure value half-way through the ascent. I think a clearer name for this quantity might be needed (if instead it is the latter, the y-axis should be reversed).**

Thank you, we will adapt the distribution of the panels in Figs. 6 and 8. In fact, reviewer 1 had the same suggestion.

As for the colors, we prefer to avoid using the colors, as this highlights that the choice of cloud microphysics has no impact. "Mean ascent height" indicates the mean pressure difference between beginning and end of ascent. This terminology follows Oertel et al. (2019) (see their Table 1); we will clarify the meaning of the word in the revised manuscript,

**Lines 216-217: Based on this result, can we say that in coarser simulations we see fewer WCB trajectories because generally ascent is less vigorous? Or are there other factors at play?**

This is an interesting idea. Within the WCB one expects a positive correlation between the strength of the WCB/number of WCB trajectories and their updraft velocity. Whether one can turn this into a causal argument of stronger ascent leading to more WCB trajectories is not clear to us, however. We therefore prefer to not include such a statement in the paper.

**Line 221: Can you elaborate on the reasons behind this statement? (slower ascent because trajectories are off-line)**

The mean slower ascent is a consequence of the offline trajectories missing (the vast majority of) the trajectories that experience embedded convection. As the convective trajectories ascent much faster than non-convective trajectories, the mean ascent period is biased high. We will include this argument in the revised manuscript.

**Line 234: I would add a rough timing of the two main periods.**

The two main periods of ascent occur at around 2016-09-23:00UTC and 2016-09-23:18UTC (see Fig. S1 in the supplemental material). The first period corresponds to the intensification of cyclone Vladiana, the

second period to its mature stage. We will including the timing in the revised manuscript.

**Fig 7: Given that $\Delta h$ is approximately proportional to - $\Delta$(logP), have you considered using log-scale in all panels (in Fig. 7 elsewhere in the manuscript) where pressure is on the x or y axis?**

We prefer to keep pressure levels in the plots. It should be easy for a reader to convert these roughly to height levels if needed.

**Line 239: I completely agree with you on the expected PV vertical profile, but this might not be familiar to all readers. Could you add a reference?**

Good idea. We will include a reference to Joos and Wernli (2012) (their Fig. 4) and Madonna et al. (2014) (their Fig. 7).

**Lines 249-250: I'm not sure I agree with this sentence. Even if the output of a 2.5km simulation is regridded to 40km, that simulation was run with the model being able to resolve processes well below 40km (although I appreciate that coarsening the output will smooth out the patterns).**

We believe this is a misunderstanding. What we mean here is that the finer grid, ascent velocity within the WCB in the 2.5 km simulation is not only stronger at small scales but also higher at the 40 km scale compared to simulations with a coarse grid-spacing. As a result, the larger-scale circulation is affected by changes at small scales that are introduced by refining the grid. This is the intention behind comparing the trajectory analysis between the simulation's "native" grid spacing and when the simulations are interpolated to the same grid spacing. This is in line with the comment of the reviewer and we will rephrase the text accordingly. For clarification, we will also include that the interpolation to the 40 km grid is done via conservative remapping.

**Figure 9: adding a line for the 2.5km simulation regridded at 40km and/or 80km would help the argument that horizontal resolution does not influence cyclone deepening.**

Thank you. In Fig. 9 a we will include the cyclone core pressure for the 2.5 km simulation with 1-moment microphysics and explicit convection for surface pressure remapped conservatively to a 1 deg x 1 deg grid. The cyclone core pressure based on the 1 deg x 1 deg evolves in a very similar manner, but the remapping leads to larger pressure values (as expected) so that the evolution closely follows the 20 and 40 km simulation. This is consistent with the cyclone deepening being largely insensitive to the grid spacing.

**Section 4.1: Would it be possible to compute PTE in a storm-relative frame of reference or, even better, to use a fully Lagrangian perspective and trace the terms onto the trajectories computed? I'm saying this as I'm thinking at ways to reduce the impact of advection and the related cancellations between terms.**

We agree that a storm-relative extension of the PTE method would be interesting. In fact, the slight mismatch between the Dp term and the actual evolution of cyclone core pressure that can be seen in Fig. 11 (top, bottom) is at least partly due to the fact that the PTE terms are not strictly computed in a storm-relative sense (see also Fink et al. (2012) for a discussion of this issue). However, the time series of Dp and the cyclone core pressure show clear structural similarities. Thus, the PTE approach in its current form is sufficient for our overall result that the deepening of Vladiana is not driven by diabatic heating.

**Line 304: I find slightly surprising that the diabatic term is not calculated explicitly but instead grouped with the residuals. Without knowing how big residuals are, I don't think it's possible making conclusions on the behaviour of the diabatic term.**

We agree that the diabatic contribution could also be computed directly from the diagnosed heating rates. While this might be slightly more accurate, we are confident it would not change the overall result that the deepening of cyclone Vladiana is not driven by diabatic processes. This is also supported by previous work of Fink et al. (2012) and Pohle (2010), who found that the diabatic term is well approximated by the residual. We will include this justification in the revised manuscript.

**References**

Binder, H., M. Boettcher, H. Joos, and H. Wernli, 2016: The Role of Warm Conveyor Belts for the Intensification of Extratropical Cyclones in Northern Hemisphere Winter. J. Atmos. Sci., 73 (10), 3997–4020, doi:10.1175/JAS-D-15-0302.1.

Blanchard, N., F. Pantillon, J.-P. Chaboureau, and J. Delanoë, 2020: Organization of convective ascents in a warm conveyor belt. Weather Clim. Dynamics, 1 (2), 617–634, doi:10.5194/wcd-1-617-2020.

Blanchard, N., F. Pantillon, J.-P. Chaboureau, and J. Delanoë, 2021: Mid-level convection in a warm conveyor belt accelerates the jet stream. Weather Clim. Dynam., 2 (1), 37–53, doi:10.5194/wcd-2-37-2021.

Fink, A. H., S. Pohle, J. G. Pinto, and P. Knippertz, 2012: Diagnosing the influence of diabatic processes on the explosive deepening of extratropical cyclones. Geophys. Res. Lett., 39 (7), L07 803, doi: 10.1029/2012GL051025.

Flack, D. L. A., G. Rivière, I. Musat, R. Roehrig, S. Bony, J. Delanoë, Q. Cazenave, and J. Pelon, 2021: Representation by two climate models of the dynamical and diabatic processes involved in the development of an explosively deepening cyclone during NAWDEX. Weather Clim. Dynam., 2 (1), 233–253, doi:10.5194/wcd-2-233-2021.

Joos, H. and H. Wernli, 2012: Influence of microphysical processes on the potential vorticity development in a warm conveyor belt: a case-study with the limited-area model COSMO. Q. J. R. Meteorol. Soc., 138 (663), 407–418, doi:https://doi.org/10.1002/qj.934.

Klocke, D., M. Brueck, C. Hohenegger, and B. Stevens, 2017: Rediscovery of the doldrums in storm-resolving simulations over the tropical Atlantic. Nature Geosci., 10, 891–896, doi: https://doi.org/10.1038/s41561-017-0005-4.

Madonna, E., H. Wernli, H. Joos, and O. Martius, 2014: Warm Conveyor Belts in the ERA-Interim Dataset (1979–2010). Part I: Climatology and Potential Vorticity Evolution. J.Climate, 27 (1), 3 – 26, doi:10.1175/JCLI-D-12-00720.1.

Mazoyer, M., et al., 2021: Microphysics Impacts on the Warm Conveyor Belt and Ridge Building of the NAWDEX IOP6 Cyclone. Mon. Weather Rev., 149 (12), 3961–3980, doi:10.1175/MWR-D-21-0061.1.

Oertel, A., M. Boettcher, H. Joos, M. Sprenger, H. Konow, M. Hagen, and H. Wernli, 2019: Convective activity in an extratropical cyclone and its warm conveyor belt – a case-study combining observations and a convection-permitting model simulation. Q. J. R. Meteorol. Soc., 145 (721), 1406–1426, doi:https://doi.org/10.1002/qj.3500.

Oertel, A., M. Boettcher, H. Joos, M. Sprenger, and H. Wernli, 2020: Potential vorticity structure of embedded convection in a warm conveyor belt and its relevance for large-scale dynamics. Weather Clim. Dynam., 1 (1), 127–153, doi:10.5194/wcd-1-127-2020.

Pohle, S., 2010: Synoptische und dynamische Aspekte tropisch-extratropischer Wechselwirkungen: Drei Fallstudien von Hitzetiefentwicklungen über Westafrika während des AMMA-Experiments 2006. Ph.D. thesis, University of Cologne, Germany; available at http://kups.ub.uni-koeln.de/volltexte/2010/3157/pdf/DissertationSusanPohle2010.pdf.

Rivière, G., M. Wimmer, P. Arbogast, J.-M. Piriou, J. Delanoë, C. Labadie, Q. Cazenave, and J. Pelon, 2021: The impact of deep convection representation in a global atmospheric model on the warm conveyor belt and jet stream during NAWDEX IOP6. Weather Clim. Dynam., 2 (4), 1011–1031, doi:10.5194/wcd-2-1011-2021.

Senf, F., A. Voigt, N. Clerbaux, A. Hünerbein, and H. Deneke, 2020: Increasing Resolution and Resolving Convection Improve the Simulation of Cloud-Radiative Effects Over the North Atlantic. J. Geophys. Res. Atmos., 125 (19), e2020JD032 667, doi:10.1029/2020JD032667.

Wimmer, M., G. Rivière, P. Arbogast, J.-M. Piriou, J. Delanoë, C. Labadie, Q. Cazenave, and J. Pelon, 2021: Diabatic processes modulating the vertical structure of the jet stream above the cold front of an extratropical cyclone: sensitivity to deep convection schemes. Weather Clim. Dynam. Discuss., 2021, 1–30, doi:10.5194/wcd-2021-76.

---

## Author Response (AR1)

**Impact of grid spacing, convective parameterization and cloud microphysics in ICON simulations of a warm conveyor belt - Letter accompanying the revised manuscript**

**Anubhav Choudhary and Aiko Voigt**

We again thank the reviewers for their encouraging evaluations and suggestions to improve our manuscript. This letter accompanies our revised manuscript. The letter is based on our response to reviewers submitted earlier. Here, after having incorporated the changes, we point more explicitly to the changes in the text. The quoted line numbers refer to the tracked changes version of the revised manuscript.

Below, we respond in detail to each of the reviewers' comments. We hope that have addressed all of their comments in a satisfying manner. The reviewers' comments are in bold, our answers are in normal font.

**Reviewer 1**

**In this paper, the representation of the warm conveyor belt (WCB) which is associated with the extratropical cyclone Vladiana in ICON simulations is discussed. The authors compare in detail the characteristics of the WCB in 18 simulations with 6 different horizontal resolutions, ranging from 80km to 2.5 km, parameterized and explicit convection, as well as 1 and 2-moment cloud microphysical parameterizations. The presented analysis is a valuable contribution to the existing literature as it is not well understood how the representation of WCBs and their impact on cyclone evolution, precipitation, the upper tropospheric flow structure and the downstream flow evolution will change with increasing resolution and the explicit treatment of convection in future weather and climate models. Although the authors do not discuss all of these implications but focus on cyclone intensification, I would recommend to accept this publication with minor revisions.**

Thank you for your interest in our work!

**Minor comments:**

**For all model resolutions you use offline trajectories which are calculated based on 1-hourly output fields. While I see the reason for that I think it would be good if you included some more critical discussion about this fact. In line 221 you say that the slower ascent is caused by the fact that you use offline instead of online trajectories (in comparison to Oertel et al., 2019). However I'm not so convinced that this is necessarily the case. By using the instantaneous value of the vertical velocity every hour in your offline trajectories it could also be that you overestimate the ascent. In case the trajectory at this point in time sees very high convective updrafts which might be relatively short lived, you assume that this high velocity lasts for one hour. This might not be so important when going to coarser resolutions but for the high resolution simulations I think it could happen. Or maybe this effects cancels in the mean over all trajectories as some of them also assume "unrealistic small updrafts speeds" over one hour? Could you add some more thoughts on this to your text?**

As noted by the reviewer, the use of offline trajectories is a pragmatic choice for the simple reason that ICON-NWP v2.1 does not include online trajectories (these have only been only implemented into a later version of ICON). We agree that the effect described by the reviewer is possible, yet Fig. S2 of the supplemental material of our original submission shows that the effect plays a minor role. Even at the finest grid

spacing and with explicit convection, we find no trajectories with hourly ascent rates above 600 hPa, and we find only very few trajectories with rates above 400 hPa. This is different from Fig. 5c of Oertel et al. (2019), for which such ascent rates occur rather frequently. As a result, the diagnosed mean ascent is slower for offline trajectories compared to online trajectories.

We have revised the manuscript accordingly to clarify this point. In particular please see lines 200f and 295f.

**I think it is interesting to see that the number of trajectories in the trajectory 2 cluster strongly increases whereas this increase is less pronounced for the other ones. I wonder what the reason for this is? Do you know where these trajectories originate in comparison to the others? Are they especially convective? Are there special mesoscale substructures e.g. in moisture and or theta values in the higher resolution simulations where these trajectories originate? Or what is the evolution of different parameters like, p, theta, q, ... along this cluster compared to the others? Maybe this is out of the scope of your study but maybe you have thoughts on this which you could include?**

We agree that this is an interesting aspect of the trajectory clusters - thank you for bringing it up. The ascent occurs in two main phases. A first bundle of trajectories starts the ascent at around 2016-09-23:00UTC; a second bundle starts the ascent at around 2019-09-23:18UTC. The increase in the number of trajectories in class 2 as the grid spacing is decreased results from an increasing number of trajectories in the second bundle. We have included this information in the revised manuscript. Please see lines 266f and 312f.

**Lines 115 ff: The synoptic description could be a little bit more precise, e.g. in line 118 you say that there are low PV values north of the British Isles, however they are low over the whole Island including Ireland... .and in Line 118: you state that the cyclone is decaying in the next 24h, however the minimum pressure in Fig. 1c is still at 985 hPa. Please adapt the description of the synoptic evolution.**

We agree and have adapted the description in the revised manuscript. Please see Sect. 2.2.

**End of chapter 4: You show that the deepening of Vladiana is not due to diabatic processes and that in this case the WCB does not contribute to the cyclone intensification. However is it possible that the WCB contributes e.g. to the formation or strengthening of PV anomalies at the cyclone's warm and cold front and that there is a difference between the different resolutions? Could a better representation of fronts and the diabatic processes in there in the higher resolution datasets lead to changed mesoscale weather phenomena, like enhanced precipitation or different precipitation patterns, changed PV anomalies associated with also differences in the wind speed?**

We agree with the reviewer that these phenomena would be interesting to study. Our interest in the paper is to test whether there is a link between the large-scale behavior of the cyclone (i.e., its deepening) and the simulation of WCB diabatic processes as a function of grid spacing and model physics, and so we prefer not to touch upon these questions in our analysis. However, we have revised the paper to mention possible WCB impacts beyond the cyclone core pressure, and to this end have specifically included the recent work of Oertel et al. (2019, 2020); Flack et al. (2021); Blanchard et al. (2020, 2021); Rivière et al. (2021) and Wimmer et al. (2021). For example, see lines 85f and 460f.

In Section 4, we now also interpret the weak WCB impact on the deepening of Vladiana using Binder et al. (2016), whose framework suggests that Vladiana is (in their terminology) a C2 cyclone with a strong WCB, but with a WCB that is far away from the cyclone center and ascents mainly along the cold front. See lines

414f.

**Style comments:**

**Line 91: ..with analysis and forecast data from the ECMWF ... why do you use forecast data here and not only analysis data?**

Note that we use data from the ECMWF operational model IFS at the highest available resolution (around 9 km) (not ERA5). The lateral boundary data is updated every 3 hours. Analysis data is available at 0 and 12 UTC, in between 3-, 6-, and 9-hr IFS forecast data are used. This is described in Senf et al. (2020). In the revised manuscript we have clarified the initial and boundary data, and replaced the reference to Klocke et al. (2017) by Senf et al. (2020). Please see lines 125f.

**E.g. line 105: make sure that the date formats are the same everywhere**

Thank you, done. Throughout the manuscript, the date format is now YYYY-MM-DDTHH.

**Line 158 end: "...to be be ..."; remove one be**

Thanks.

**Figure 2 caption: trajectory 3 (red), should be "yellow"?**

Yes, indeed. Corrected.

**Figure 5 caption, 2. Line: ..., with given by the right x-axis $\rightarrow$ right y-axis**

Thanks and corrected.

**Figure 5: Panels a and b should be horizontally aligned, or maybe even remove panel b as not much is changing and only mention it in the text**.

We have realigned the figure. Yet, we prefer to keep the panel b with the simulations using the 2-moment microphysics to explicitly show that microphysics play a minor role.

**Figure 6: Align as 2X2 figure, not all 4 in a column**

Good suggestion, thank you and adapted accordingly.

**Line 262: Section title: "4 Synoptic evolution of cyclone Vladiana ..." $\rightarrow$ "4 Pressure evolution of ..."**

Thanks, we now use "pressure evolution" in the revised manuscript to make it clear that we only consider the cyclone core pressure in this section.

**Figure 8: align panels horizontally and vertically**

Yes - panels are not realigned.

**Line 293: integral $\rightarrow$ integral**

Thanks, corrected.

**In the reference list there are still many typos (e.g. capital letters, " " signs, ...), please correct.**

Thank you, and please accept our apologies for not having checked the reference list carefully. We have carefully checked and corrected the reference list in the revised manuscript. (not highlighted in the tracked changes version as latexdiff is unable to catch these changes in the bbtex file)

**Reviewer 2**

**This manuscript contains a nice, detailed, and insightful analysis of 18 ICON simulations of cyclone Vladiana. The authors focus on the dynamics of the warm conveyor belt and on its relationship with model horizontal resolution, convection settings and microphysics scheme. This is a well-written piece of research that could surely be a welcome contribution to the field. However, I have some reservations on the application of the PTE method (see the two relevant comments at the end of this document) and, as such, I don't think the third research question (see Conclusions) has been fully answered as of now. Apart from this, all other comments are minor and, in some cases, I'm just requesting to expand some discussions and clarify some statements. Overall, I think this paper would be worth of publication once the comments are addressed.**

Thank you for your interest in our work!

General comments:

**I find the title slightly vague and I don't think it does justice to the novelty of your analysis. I appreciate that for space reasons it won't be possible to explicitly mention model resolution, convective parametrization and microphysics scheme, but I would suggest you try highlighting the analysis performed on these ICON simulations.**

Thank you, we have changed the title to "Impact of grid spacing, convective parameterization and cloud microphysics in ICON simulations of a warm conveyor belt" (note the slight change to our earlier response letter). The new title highlights the analysis approach. The words "cloud diabatic" and "North Atlantic cyclone Vladiana" will be cut as they are not essential, allowing for an in fact shorter title.

**I would consider replacing "higher/highest" and "lower/lowest" resolution with "smoother/smoothest" and "coarser/coarsest", respectively. Given that grid spacing decreases with finer ("higher") resolution, those two terms might be misleading (the same applies to all the "increasing/decreasing resolution" instances).**

We agree that this can be confusing. We have replaced "resolution" by "grid spacing" throughout the manuscript and adapted the sentences accordingly.

**DOIs seem to be missing from most of the references included in this manuscript. It would be helpful for reviewers & readers to add them, so the referenced research can be accessed more quickly.**

Corrected - dois are now included for all references.

Comments regarding the abstract:

**Lines 7-8: "the resolution" here could be replaced by "that" in both instances, making the sentence more concise.**

Agreed and adapted. See lines 9 and 8.

**Lines 13: "With increasing resolution". I would specify exactly what are the two resolutions that show a 3-time increase in ascent velocity. Also, if there's space, I would include a brief statement on the saturation of those effects when moving to resolutions finer than 10 km.**

Agreed and adapted. See line 16 and lines 20f.

**Lines 17-18: "on the one hand" and "on the other hand" are not really needed here.**

We have removed "on the one hand" and "on the other hand".

**Introduction:**

**There is a sort of jump between the first two paragraphs, describing the WCB, and the following one, starting with "Despite decades of model development...". I would suggest improving the link between the different topics presented and/or to try change the order of paragraphs and see if the readability improves.**

Thank you, we will try to rephrase these paragraphs to improve the flow.

**Lines 23-24: "Most of the diabatic processes occur within coherent streams of ascending air known as warm conveyor belts". This is a bold claim and I don't think the papers referenced justify it. I understand the reasoning behind it, but I would still like it to be rephrased (or properly justified) as substantial diabatic processes can be found in many other parts of the cyclone, even outside the warm sector.**

Agreed. In fact, we did not mean to imply that diabatic processes are limited to the WCB and have rephrased accordingly. See lines 30f.

**Lines 44-45: It might be better saying "... in which horizontal grid spacing is reduced to a few kilometers ..." (or something similar, should you prefer taking into account spectral models).**

We agree and have rephrased as suggested. See line 61.

**Method:**

**Lines 89-90: Could you give some motivation as to why it is ok that the model follows a "tropical Atlantic setup"?**

Sorry for the confusing formulation. What we meant is that our simulations use essentially the same model setup (apart from the domain location) as the (sub)tropical North Atlantic simulations of Klocke et al. (2017). However, the formulation in the original manuscript was vague and misleading. We have removed the phrase regarding the "tropical Atlantic setup" and we now instead refer to Senf et al. (2020). Senf et al. (2020) analyzed the same set of North-Atlantic simulations that we study in our paper, and provided a detailed description of the model setup. See lines 119f.

**Line 91: Consider moving the domain description here (or motivate why it should stay at lines 106-107).**

Thank you, we have moved the text. See lines 122-125.

**Line 94: Did you consider running half-explicit convection (i.e., with only shallow or deep parameterizations disabled)? I'm not asking you to run anymore simulations, I was just wondering if you thought such as setup could be useful (possibly in future works?)**

One of us (AV) has performed and analyzed simulations with active shallow but disabled deep convective parameterization. These simulations are included in Senf et al. (2020), who evaluated them in terms of North Atlantic clouds and top-of-atmosphere radiative effects. The simulations are not included here for the pragmatic reason that they were run only after the technical aspects of the WCB trajectory analysis was already finished.

**Line 105: Could you briefly specify here how what the 18 simulations are? (I don't think a table is needed, a sentence in the text should be enough).**

Yes, of course. We have added a corresponding sentence in the revised manuscript. See lines 149f.

**Figure 1: I would replace the PV colour scale with a more intuitive (and colour-blind friendly) one. I understand the emphasis on the 2 PVU value, but I think it could just by highlighted with a red contour line, while using a more logical colour scale for the shading. Also, I would replace "EQ" with "0W" on the x axis.**

We wish to keep the PV color scale in its current form as it mimics the color scale of Fig. 2 d-f of Oertel et al. (2020). We now make this choice clear in the caption. And yes, on the x-axis "EQ" should of course be "0W"!

**Line 135-137: "from every grid point": this would suggest that simulations with finer resolutions have more trajectory starting points, whereas line 137 indicates that this is not the case as "the seeding points are based on the 20 km simulation. For all resolutions...". I assume the latter is true but a clarification is needed, for example moving this second quote up to line 135 (and, if my assumption is wrong, "number" needs to be replaced with "proportion" at line 9).**

Thank you, the formulation "from every grid point" was indeed confusing. We have clarified this in the revised manuscript. In particular, we now mention that the number of seeded trajectories is the same for all simulations, and in fact is 395,825. See lines 187f.

**Line 147: "As a result, ..." could you clarify why this point about slantwise convection is a consequence of using off-line rather than on-line trajectories?**

The slantwise character follows from two points. First, in the coarse-grid simulations the convection scheme is active and the grid spacing is too large to allow for convective updrafts by the resolved flow. Second, the fine-grid simulations allow for convective updrafts – however, the updrafts are largely missed as they are short-lived and the wind-field is sampled only hourly. Fig. S2 in the supplemental material illustrates the largely slantwise character of the ascent: the region of strong updrafts is unpopulated, signaling that the ascent is slantwise. Yet, we realize now that we did not refer to Fig. S2 in the original submission. In the revised manuscript we now include the above discussion and a reference to Fig. S2. See lines 200f.

**2.4 Diabatic Heating: Are these tendencies computed on-line by the model?**

Yes, the diabatic tendencies are diagnosed online. Now stated in line 214.

**WCB trajectories:**

**Lines 177-178: Is it possible to be more quantitative here? (giving more context to the following split into subsets, and Figure 4)**

Yes, good idea. We have included the information in lines 237f and now link the text to Figs. 4 and 5.

**Line 182: I would say "more" than 10 times, looking particularly at Figure 3.**

Actually, we have kept the text. More than 10 times is only correct for the 2-moment runs but not for the 1-moment runs.

**Lines 185-187: Again, could you make this statement more quantitative?**

We have made the statement more quantitative. In essence, the number of trajectories increases by 30% between the 10 km and 2.5 km grid spacing for parameterized convection, but does not change systematically with grid spacing for explicit convection. See lines 245f.

**Figure 4: How are the boundaries of the rectangular domains defined? Is there any motivation to not just stop at an "anticyclonic vs cyclonic" decomposition?**

The boundaries were chosen based on two reasons. First, we wanted to separate cyclonic and anticyclonic trajectories. Then, given the high number of anticyclonic trajectories and their clear differences in final location, we further separated anticyclonic trajectories into 3 classes. The distinct behavior of class 4, which turn southward very quickly, was a particular motivation for this additional separation.

**Figure 5: I would consider changing this figure to a 4-panel one, to separate total and subclasses counts. Other solutions are welcome, as long as they make it clearer to interpret (I find having all and subclasses together, and on two different scales, slightly confusing).**

Thank you. However, we prefer to keep the figure with all trajectories and the subclasses in the same panel. This makes it easier to compare how the grid-spacing dependence of the number of all trajectories relates to the subclasses.

**Figure 6: I suggest making this a 2x2-panel figure (a similar consideration applies to Fig 8, which could become a 3x2 or 2x3 one). I think that using colours would help differentiating between the various points. Also, I suppose that "mean ascent height" indicates the mean pressure difference between beginning and end of the ascent, rather than the pressure value half-way through the ascent. I think a clearer name for this quantity might be needed (if instead it is the latter, the y-axis should be reversed).**

Thank you, we have adapted the distribution of the panels in Figs. 6 and 8. In fact, reviewer 1 had the same suggestion.

As for the colors, we prefer to avoid using colors, as this highlights that the choice of cloud microphysics has no impact. "Mean ascent height" indicates the mean pressure difference between beginning and end of ascent. This terminology follows Oertel et al. (2019) (see their Table 1); we have clarified the meaning of the word in the revised manuscript. See line 275.

**Lines 216-217: Based on this result, can we say that in coarser simulations we see fewer WCB trajectories because generally ascent is less vigorous? Or are there other factors at play?**

This is an interesting idea. Within the WCB one expects a positive correlation between the strength of the WCB/number of WCB trajectories and their updraft velocity. Whether one can turn this into a causal argument of stronger ascent leading to more WCB trajectories is not clear to us, however. We therefore prefer to not include such a statement in the paper.

**Line 221: Can you elaborate on the reasons behind this statement? (slower ascent because trajectories are off-line)**

The mean slower ascent is a consequence of the offline trajectories missing (the vast majority of) the trajectories that experience embedded convection. As the convective trajectories ascent much faster than non-convective trajectories, the mean ascent period is biased high. We have included this argument in the revised manuscript in lines 295f.

**Line 234: I would add a rough timing of the two main periods.**

The two main periods of ascent occur at around 2016-09-23:00UTC and 2016-09-23:18UTC (see Fig. S1 in the supplemental material). The first period corresponds to the intensification of cyclone Vladiana, the second period to its mature stage. We have included the timing in the revised manuscript in lines 267f.

**Fig 7: Given that $\Delta h$ is approximately proportional to - $\Delta$(logP), have you considered using log-scale in all panels (in Fig. 7 elsewhere in the manuscript) where pressure is on the x or y axis?**

We prefer to keep pressure levels in the plots. It should be easy for a reader to convert these roughly to height levels if needed.

**Line 239: I completely agree with you on the expected PV vertical profile, but this might not be familiar to all readers. Could you add a reference?**

Good idea. We have added references to Joos and Wernli (2012) (their Fig. 4) and Madonna et al. (2014) (their Fig. 7) in line 318.

**Lines 249-250: I'm not sure I agree with this sentence. Even if the output of a 2.5km simulation is regridded to 40km, that simulation was run with the model being able to resolve processes well below 40km (although I appreciate that coarsening the output will smooth out the patterns).**

We believe this is a misunderstanding. What we mean here is that the finer grid, ascent velocity within the WCB in the 2.5 km simulation is not only stronger at small scales but also higher at the 40 km scale compared to simulations with a coarse grid-spacing. As a result, the larger-scale circulation is affected by changes at small scales that are introduced by refining the grid. This is the intention behind comparing the trajectory analysis between the simulation's "native" grid spacing and when the simulations are interpolated to the same grid spacing. This is in line with the comment of the reviewer and we have rephrase the text accordingly in lines 330f. For clarification, we now also state that the interpolation to the 40 km grid is done via conservative remapping.

**Figure 9: adding a line for the 2.5km simulation regridded at 40km and/or 80km would help the argument that horizontal resolution does not influence cyclone deepening.**

Thank you. In Fig. 9 a we have included the cyclone core pressure for the 2.5 km simulation with 1-moment microphysics and parameterized convection for surface pressure remapped conservatively to a 1 deg x 1 deg grid. The cyclone core pressure based on the 1 deg x 1 deg evolves in a very similar manner, but the remapping leads to larger pressure values (as expected) so that the evolution closely follows the 20 and 40 km simulation. This is consistent with the cyclone deepening being largely insensitive to the grid spacing.

**Section 4.1: Would it be possible to compute PTE in a storm-relative frame of reference or, even better, to use a fully Lagrangian perspective and trace the terms onto the trajectories computed? I'm saying this as I'm thinking at ways to reduce the impact of advection and the related cancellations between terms.**

We agree that a storm-relative extension of the PTE method would be interesting. In fact, the slight mismatch between the Dp term and the actual evolution of cyclone core pressure that can be seen in Fig. 11 (top, bottom) is at least partly due to the fact that the PTE terms are not strictly computed in a storm-relative sense (see also Fink et al. (2012) for a discussion of this issue). However, the time series of Dp and the cyclone core pressure show clear structural similarities. Thus, the PTE approach in its current form is sufficient for our overall result that the deepening of Vladiana is not driven by diabatic heating.

**Line 304: I find slightly surprising that the diabatic term is not calculated explicitly but instead grouped with the residuals. Without knowing how big residuals are, I don't think it's possible making conclusions on the behaviour of the diabatic term.**

We agree that the diabatic contribution could also be computed directly from the diagnosed heating rates. While this might be slightly more accurate, we are confident it would not change the overall result that the deepening of cyclone Vladiana is not driven by diabatic processes. This is also supported by previous work of Fink et al. (2012) and Pohle (2010), who found that the diabatic term is well approximated by the residual. We have included this justification in the revised manuscript in line 396.

**Other changes**

In the revised paper, we have include references to recent work on WCBs and extratropical cyclones from the NAWDEX campaign. This includes the work of Blanchard et al. (2020, 2021); Flack et al. (2021); Wimmer et al. (2021); Rivière et al. (2021) and Mazoyer et al. (2021).

We have reworked the captions of the figures in the supplementary material.

We have also done editorial changes to the manuscript that we hope improve its readability. These changes are not specifically listed here but are included in the tracked changes version.

**References**

Binder, H., M. Boettcher, H. Joos, and H. Wernli, 2016: The Role of Warm Conveyor Belts for the Intensification of Extratropical Cyclones in Northern Hemisphere Winter. J. Atmos. Sci., 73 (10), 3997–4020, doi:10.1175/JAS-D-15-0302.1.

Blanchard, N., F. Pantillon, J.-P. Chaboureau, and J. Delanoë, 2020: Organization of convective ascents in a warm conveyor belt. Weather Clim. Dynamics, 1 (2), 617–634, doi:10.5194/wcd-1-617-2020.

Blanchard, N., F. Pantillon, J.-P. Chaboureau, and J. Delanoë, 2021: Mid-level convection in a warm conveyor belt accelerates the jet stream. Weather Clim. Dynam., 2 (1), 37–53, doi:10.5194/wcd-2-37-2021.

Fink, A. H., S. Pohle, J. G. Pinto, and P. Knippertz, 2012: Diagnosing the influence of diabatic processes on the explosive deepening of extratropical cyclones. Geophys. Res. Lett., 39 (7), L07 803, doi:10.1029/2012GL051025.

Flack, D. L. A., G. Rivière, I. Musat, R. Roehrig, S. Bony, J. Delanoë, Q. Cazenave, and J. Pelon, 2021: Representation by two climate models of the dynamical and diabatic processes involved in the development of an explosively deepening cyclone during NAWDEX. Weather Clim. Dynam., 2 (1), 233–253, doi:10.5194/wcd-2-233-2021.

Joos, H. and H. Wernli, 2012: Influence of microphysical processes on the potential vorticity development in a warm conveyor belt: a case-study with the limited-area model COSMO. Q. J. R. Meteorol. Soc., 138 (663), 407–418, doi:https://doi.org/10.1002/qj.934.

Klocke, D., M. Brueck, C. Hohenegger, and B. Stevens, 2017: Rediscovery of the doldrums in storm-resolving simulations over the tropical Atlantic. Nature Geosci., 10, 891–896, doi:https://doi.org/10.1038/s41561-017-0005-4.

Madonna, E., H. Wernli, H. Joos, and O. Martius, 2014: Warm Conveyor Belts in the ERA-Interim Dataset (1979–2010). Part I: Climatology and Potential Vorticity Evolution. J.Climate, 27 (1), 3 – 26, doi:10.1175/JCLI-D-12-00720.1.

Mazoyer, M., et al., 2021: Microphysics Impacts on the Warm Conveyor Belt and Ridge Building of the NAWDEX IOP6 Cyclone. Mon. Weather Rev., 149 (12), 3961–3980, doi:10.1175/MWR-D-21-0061.1.

Oertel, A., M. Boettcher, H. Joos, M. Sprenger, H. Konow, M. Hagen, and H. Wernli, 2019: Convective activity in an extratropical cyclone and its warm conveyor belt – a case-study combining observations and a convection-permitting model simulation. Q. J. R. Meteorol. Soc., 145 (721), 1406–1426, doi:https://doi.org/10.1002/qj.3500.

Oertel, A., M. Boettcher, H. Joos, M. Sprenger, and H. Wernli, 2020: Potential vorticity structure of embedded convection in a warm conveyor belt and its relevance for large-scale dynamics. Weather Clim. Dynam., 1 (1), 127–153, doi:10.5194/wcd-1-127-2020.

Pohle, S., 2010: Synoptische und dynamische Aspekte tropisch-extratropischer Wechselwirkungen: Drei Fallstudien von Hitzetiefentwicklungen über Westafrika während des AMMA-Experiments 2006. Ph.D. thesis, University of Cologne, Germany; available at http://kups.ub.uni-koeln.de/volltexte/2010/3157/pdf/DissertationSusanPohle2010.pdf.

Rivière, G., M. Wimmer, P. Arbogast, J.-M. Piriou, J. Delanoë, C. Labadie, Q. Cazenave, and J. Pelon, 2021: The impact of deep convection representation in a global atmospheric model on the warm conveyor belt and jet stream during NAWDEX IOP6. Weather Clim. Dynam., 2 (4), 1011–1031, doi:10.5194/wcd-2-1011-2021.

Senf, F., A. Voigt, N. Clerbaux, A. Hünerbein, and H. Deneke, 2020: Increasing Resolution and Resolving Convection Improve the Simulation of Cloud-Radiative Effects Over the North Atlantic. J. Geophys. Res. Atmos., 125 (19), e2020JD032 667, doi:10.1029/2020JD032667.

Wimmer, M., G. Rivière, P. Arbogast, J.-M. Piriou, J. Delanoë, C. Labadie, Q. Cazenave, and J. Pelon, 2021: Diabatic processes modulating the vertical structure of the jet stream above the cold front of an extratropical cyclone: sensitivity to deep convection schemes. Weather Clim. Dynam. Discuss., 2021, 1–30, doi:10.5194/wcd-2021-76.

---

## Author Response (AR2)

**Impact of grid spacing, convective parameterization and cloud microphysics in ICON simulations of a warm conveyor belt- Letter accompanying the corrected manuscript**

**Anubhav Choudhary and Aiko Voigt**

We again thank the reviewer for their valuable suggestions which helped in improving our manuscript to its final form and recommending its acceptance. This letter accompanies our manuscript after incorporating technical correction as suggested by Referee #2. Here, after having incorporated the changes, we point more explicitly to the changes in the text. The reviewers' comments are in bold, our answers are in normal font.

**Referee#2**

**Many thanks to the authors for considering my comments and revising the manuscript accordingly. I am satisfied with the revisions and I am happy for the manuscript to be accepted in its current form, after one small remaining comment is addressed, concerning Figure 6.**

**My concern is that a quantity called "height" cannot have "hPa" as units, even though I appreciate the need for consistency with the reference cited. I suggest to rename the quantity as "mean ascent deltaP" or similar (or even just "mean ascent") and then explain in the text that this quantity is analogous/equivalent to that described in the reference cited.**

Thank you for appreciating our effort we put in revision and recommending its acceptance.

We agree that the term 'height' is a bit misleading in this context. Therefore, as suggested, we have renamed the quantity as 'mean ascent'. The title of Figure 6c is modified accordingly. Also, in the corresponding text, all the instances of 'mean ascent height' are now changed to 'mean ascent' including a modification in line- 223, where mean ascent is defined. As suggested, in line 224, we have mentioned that this quantity is equivalent to that used by Oertel et al. (2019) in their Table 1.